# Latent Denoising Makes Good Tokenizers

**Jiawei Yang**[1]    **Tianhong Li**[2]    **Lijie Fan**[3*]    **Yonglong Tian**[4†]    **Yue Wang**[1]

[1]USC    [2]MIT CSAIL    [3]Google DeepMind    [4]OpenAI

## Abstract

Despite their fundamental role, it remains unclear what properties could make tokenizers more effective for generative modeling. We observe that modern generative models share a conceptually similar training objective—reconstructing clean signals from corrupted inputs, such as signals degraded by Gaussian noise or masking—a process we term *denoising*. Motivated by this insight, we propose aligning tokenizer embeddings directly with the downstream denoising objective, encouraging latent embeddings that remain reconstructable even under significant corruption. To achieve this, we introduce the Latent Denoising Tokenizer ($l$-DeTok), a simple yet highly effective tokenizer trained to reconstruct clean images from latent embeddings corrupted via interpolative noise or random masking. Extensive experiments on class-conditioned (ImageNet $256 \times 256$ and $512 \times 512$) and text-conditioned (MSCOCO) image generation benchmarks demonstrate that our $l$-DeTok consistently improves generation quality across *six* representative generative models compared to prior tokenizers. Our findings highlight denoising as a fundamental design principle for tokenizer development, and we hope it could motivate new perspectives for future tokenizer design. Code is available at: https://github.com/Jiawei-Yang/DeTok.

## 1 Introduction

Modern visual generative models commonly operate on compact *latent embeddings*, produced by tokenizers, to circumvent the prohibitive complexity of pixel-level modeling (Rombach et al., 2022; Peebles & Xie, 2023; Chang et al., 2022; Li et al., 2024a). Current tokenizers are typically trained as standard variational autoencoders (Kingma & Welling, 2014), primarily optimizing for pixel-level reconstruction. Despite their critical influence on downstream generative quality, it remains unclear what properties enable more effective tokenizers for generation. As a result, tokenizer development has lagged behind recent rapid advances in generative model architectures.

In this work, we ask: *What properties can make visual tokenizers more effective for generative modeling?* We observe that modern generative models, despite methodological differences, share a conceptually similar training objective—reconstructing original signals from corrupted ones. For instance, diffusion models remove diffusion-induced noise to recover clean signals (Ho et al., 2020; Peebles & Xie, 2023), while autoregressive models reconstruct complete sequences from partially observed contexts (Li et al., 2024a; Chang et al., 2022), analogous to removing "masking noise" (He et al., 2022; Chen et al., 2025c; Devlin et al., 2019). We collectively refer to these reconstruction-from-deconstruction[1] processes as *denoising*.

This unified *denoising* perspective of modern generative models suggests that effective visual tokenizers for these models should produce latent embeddings that are reconstructable even under significant corruption. Such embeddings naturally align with the denoising objectives of downstream generative models, facilitating their training and subsequently enhancing their generation quality.

Motivated by this insight, we propose to train tokenizers as latent denoising autoencoders, termed as $l$-DeTok. Specifically, we corrupt latent embeddings via *interpolative noise*, generated by interpolating original embeddings with Gaussian noise. The tokenizer decoder is then trained to reconstruct

---

[*]Advisory-only

[†]Work done prior to joining OpenAI

[1]We use the terms "deconstruction" and "corruption" interchangeably throughout the paper.

clean images from these heavily noised latent embeddings. Additionally, we explore random masking, akin to masked autoencoders (MAE) (He et al., 2022), as an alternative and optional form of deconstruction and find it similarly effective.

Conceptually, these deconstruction-reconstruction strategies encourage latent embeddings to be robust, stable, and easily reconstructable under strong corruption, aligning with the downstream denoising tasks central to generative models. Indeed, our experiments show that stronger noise used in our $l$-DeTok training typically yields better downstream generative performance.

We demonstrate the effectiveness and generalizability of $l$-DeTok across *six representative generative models*, including non-autoregressive (DiT (Peebles & Xie, 2023), SiT (Ma et al., 2024), LightningDiT (Yao & Wang, 2025)) and autoregressive models (MAR (Li et al., 2024a), RasterAR, RandomAR (Li et al., 2024a; Pang et al., 2025; Yu et al., 2024)) on the ImageNet generation benchmark. By adopting our tokenizer—without modifying model architectures—we push the limits for MAR models (Li et al., 2024a), improving FID from 2.31 to 1.55 for MAR-B, matching the performance of the original huge-sized MAR (1.55). For MAR-L, FID improves from 1.78 to 1.35. Importantly, these gains come *without* semantics distillation, thus avoiding reliance on visual encoders pretrained at a far larger scale (Oquab et al., 2024; Radford et al., 2021).

In summary, our work demonstrates a simple yet crucial insight: explicitly incorporating denoising objectives into tokenizer training significantly enhances their effectiveness for generative modeling since it is downstream task-aligned. We hope this perspective will stimulate new research directions in tokenizer design and accelerate future advances in generative modeling.

## 2 RELATED WORK

**Representation learning in visual recognition.** Representation learning has been a decades-long pursuit in visual recognition, aiming to obtain transferable embeddings that generalize across diverse downstream tasks (Bengio et al., 2013). At the core of these approaches lies a foundational principle: pre-training should encourage representations to encode the information most relevant to downstream tasks. This principle has inspired the design of diverse and effective pretext tasks, such as instance discrimination (He et al., 2020; Chen et al., 2020b), self-distillation (Grill et al., 2020; Caron et al., 2021; Oquab et al., 2024), and masked-image reconstruction (He et al., 2022; Oquab et al., 2024; Zhou et al., 2022), each aligning representations with downstream utility. These insights motivate us to explore tokenizer embeddings that align with downstream generative tasks.

**Visual tokenizers for generative modeling.** Modern generative models rely on tokenizers to encode images into compact latent embeddings, significantly reducing computational complexity compared to pixel-level modeling. While conventional tokenizers optimize pixel reconstruction with KL-regularization, recent approaches (Chen et al., 2025a; Yao & Wang, 2025; Chen et al., 2025b; Li et al., 2025) have advocated semantics distillation from powerful pretrained vision models (Oquab et al., 2024; Radford et al., 2021). Yet, in many domains (*e.g.*, video, audio, 3D/4D), such encoders may not exist. In contrast, our $l$-DeTok learns good tokenizers without these dependencies, and it could be complementary to approaches such as $\epsilon$-VAE, which replaces the deterministic VAE decoder with a denoising diffusion process guided by encoder latents (Zhao et al., 2025), and residual vector-quantized tokens, which support high-fidelity yet efficient discrete diffusion generation via collective token prediction (Kim et al., 2025).

**Generative modeling frameworks.** Generative modeling frameworks broadly include autoregressive (AR) (Chen et al., 2020a; Esser et al., 2021; Chang et al., 2022; Li et al., 2024a; Tian et al., 2025; Pang et al., 2025; Yu et al., 2024; Tschannen et al., 2024) and diffusion-based non-autoregressive (non-AR) methods (Ho et al., 2020; Rombach et al., 2022; Peebles & Xie, 2023; Ma et al., 2024; Yao & Wang, 2025). AR models predict latent tokens sequentially conditioned on partial contexts, while non-AR models jointly generate tokens via iterative refinement, typically using diffusion (Ho et al., 2020; Sohl-Dickstein et al., 2015) or flow-based processes (Lipman et al., 2023; Esser et al., 2024). Despite methodological differences, both paradigms critically rely on high-quality latent embeddings from upstream tokenizers (Hansen-Estruch et al., 2025). Thus, improving tokenizer embeddings is fundamental for enhancing generative performance across diverse frameworks.

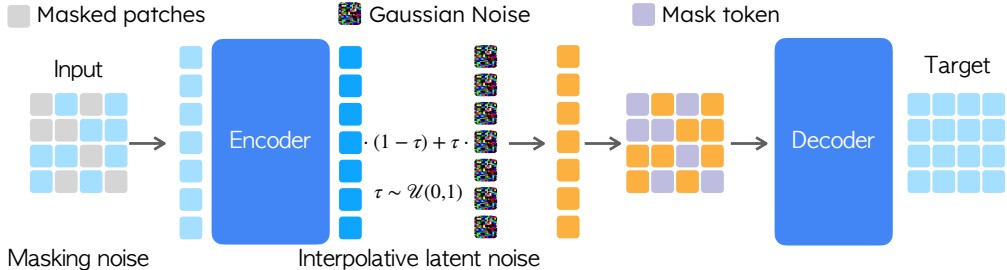

Figure 1: **Our latent denoising tokenizers ($l$-DeTok) framework.** During tokenizer training, we randomly mask input patches (*masking noise*) and interpolate encoder-produced latent embeddings with Gaussian noise (*interpolative latent noise*). The decoder processes these deconstructed latents and mask tokens to reconstruct the original images in pixels. We refer to this process as *denoising*. When serving as a tokenizer for downstream generative models, both noises are disabled.

## 3 METHOD

Our goal is to design visual tokenizers that are more effective for generative modeling compared to standard autoencoders trained for pixel reconstruction. This section first revisits the core training objective shared by all modern generative models, *i.e.*, denoising, to motivate our design, and then details our latent denoising tokenizers ($l$-DeTok).

### 3.1 PRELIMINARIES OF GENERATIVE MODELING

Modern generative frameworks can be primarily divided into non-autoregressive (non-AR) and autoregressive (AR) paradigms. Despite their methodological differences, both paradigms aim to gradually reconstruct the original representations from deconstructed ones.

**Non-autoregressive generative models.** Non-autoregressive models, exemplified by diffusion (Ho et al., 2020; Peebles & Xie, 2023) and flow-matching methods (Lipman et al., 2023), learn to iteratively refine latent representations deconstructed by controlled noise. Given a latent representation of input $\mathbf{X}_0$, the forward noising process progressively corrupts these latents into $\mathbf{X}_t$:

$$\mathbf{X}_t = a(t)\,\mathbf{X}_0 + b(t)\,\boldsymbol{\epsilon}_t, \quad \boldsymbol{\epsilon}_t \sim \mathcal{N}(\mathbf{0}, \mathbf{I}), \tag{1}$$

where $a(t)$ and $b(t)$ are noise schedules. Generative models are trained to revert this deconstruction:

$$\mathcal{L}(\boldsymbol{\theta}) = \mathbb{E}_{\mathbf{X}, \boldsymbol{\epsilon}, t}\left[\left\|\epsilon_{\boldsymbol{\theta}}(\mathbf{X}_t, t) - \boldsymbol{\epsilon}_t\right\|^2\right], \tag{2}$$

where $\epsilon_{\boldsymbol{\theta}}$ is a learnable noise estimator parameterized by $\boldsymbol{\theta}$. Essentially, non-AR diffusion models learn to *reconstruct original latents from intermediate latents deconstructed by noise*.

**Autoregressive generative models.** Autoregressive approaches factorize image generation into a sequential prediction problem. Given an ordered sequence of latent tokens $\{\mathbf{x}^1, \ldots, \mathbf{x}^N\}$, AR methods factorize the joint distribution as:

$$p_{\boldsymbol{\theta}}(\mathbf{x}) = \prod_{i=1}^{N} p_{\boldsymbol{\theta}}(\mathbf{x}^i | \mathbf{x}^1, \ldots, \mathbf{x}^{i-1}), \tag{3}$$

where $\mathbf{x}^i$ denotes the latent tokens generated at step $i$. Recent generalized AR variants extend this framework to arbitrary generation orders (Li et al., 2024a; Yu et al., 2024; Pang et al., 2025) or set-wise prediction strategies (Tian et al., 2025; Ren et al., 2025). Nonetheless, the fundamental training objective remains consistent: reconstructing full sequences from partially observed—or equivalently, partially *masked*—contexts. In other words, AR models learn to *reconstruct original latents from intermediate latents deconstructed by masking*.

### 3.2 LATENT DENOISING TOKENIZERS

Motivated by the discussions above, we propose latent denoising tokenizer ($l$-DeTok), a simple tokenizer trained by reconstructing original images from deconstructed latent representations. This deconstruction-reconstruction design aligns with the denoising tasks employed by modern generative models. Figure 1 shows an overview of our method. We detail each component next.

**Overview.** Our tokenizer follows an encoder-decoder architecture based on Vision Transformers (ViT) (Dosovitskiy et al., 2021). Input images are divided into non-overlapping patches, linearly projected into embedding vectors, and added with positional embeddings. During training, we deconstruct these embeddings using two complementary strategies: (i) injecting noise in latent embeddings and (ii) randomly masking image patches. The decoder reconstructs original images from these deconstructed embeddings. This strategy encourages easy-to-reconstruct latent embeddings under heavy corruption, aiming to simplify downstream denoising tasks in generative models.

**Noising as deconstruction.** Our core idea is to deconstruct latent embeddings by noise. Specifically, given latent embeddings $\mathbf{x}$ from the encoder, we *interpolate* them with Gaussian noise as follows:

$$\mathbf{x}' = (1 - \tau)\mathbf{x} + \tau\boldsymbol{\varepsilon}(\gamma), \quad \text{where} \quad \boldsymbol{\varepsilon}(\gamma) \sim \gamma \cdot \mathcal{N}(\mathbf{0}, \mathbf{I}), \quad \tau \sim \mathcal{U}(0, 1). \tag{4}$$

Here, the scalar $\gamma$ controls noise standard deviation, and the factor $\tau$ specifies noise level. Critically, this *interpolative* strategy differs from the conventional additive noise, *i.e.*, $\mathbf{x}' = \mathbf{x} + \tau\boldsymbol{\varepsilon}$, employed by standard VAEs (Pu et al., 2016), as it ensures latents can be effectively and heavily corrupted when the noise level $\tau$ is high. Moreover, random sampling of $\tau$ encourages latents to remain robust across diverse corruption levels. At inference time, latent noising is disabled ($\tau = 0$).

**Masking as deconstruction.** We further generalize our denoising perspective by interpreting masking as another form of latent deconstruction (Chen et al., 2025c). Inspired by masked autoencoders (MAE) (He et al., 2022), we randomly mask a subset of image patches. Different from MAE, we use a random masking ratio. Concretely, given an input image partitioned into patches, we mask a random subset, where the masking ratio $m$ is sampled from a slightly biased uniform distribution:

$$m = \max(0, \mathcal{U}(-0.1, M)), \tag{5}$$

where $\mathcal{U}(-0.1, M)$ denotes a uniform distribution on $[-0.1, M]$. The slight bias towards zero reduces the distribution gap between training and inference (no masking). The encoder processes only the visible patches, and masked positions are represented by shared learnable `[MASK]` tokens at the decoder input. At inference time, all patches are visible ($m = 0$).

**Training objectives.** Our tokenizer is trained to reconstruct the original images from corrupted latent embeddings. The training objective follows established practice (Rombach et al., 2022; Esser et al., 2021; Yu et al., 2025a), combining pixel-wise mean-squared-error (MSE), latent-space KL-regularization (Pu et al., 2016), perceptual losses (VGG- and ConvNeXt-based as in Yu et al. (2025a)), and an adversarial GAN objective (Goodfellow et al., 2014):

$$\mathcal{L}_{\text{total}} = \mathcal{L}_{\text{MSE}} + \lambda_{\text{KL}}\mathcal{L}_{\text{KL}} + \lambda_{\text{percep}}\mathcal{L}_{\text{percep}} + \lambda_{\text{GAN}}\mathcal{L}_{\text{GAN}}, \tag{6}$$

where each $\lambda$ controls the contribution of the corresponding loss component.

## 4 IMPLEMENTATION

We briefly summarize implementation details here, including datasets, evaluation metrics, and model training procedures. Due to limited space, we defer comprehensive details to Sections A.1 and A.2.

**Dataset and metrics.** We primarily experiment with class-conditioned image generation on the ImageNet dataset (Russakovsky et al., 2015) at $256 \times 256$ and $512 \times 512$ resolutions, and further evaluate text-to-image generation on the MS-COCO dataset (Lin et al., 2014). For evaluation, we report Fréchet Inception Distance (FID), Inception Score (IS), precision, and recall.

**Tokenizer baselines.** We benchmark our tokenizer against a diverse set of tokenizers: (i) MAR-VAE from MAR (Li et al., 2024a), trained on ImageNet using the implementation from Esser et al. (2021); (ii) VA-VAE (Yao & Wang, 2025), aligning latent embeddings with DINOv2 features; (iii) MAE-Tok (Chen et al., 2025a), distilling HOG, DINOv2 (Oquab et al., 2024), and CLIP (Radford et al., 2021) features through auxiliary decoders; (iv) SD-VAE and SD3-VAE from Stable-Diffusion (Rombach et al., 2022; Esser et al., 2024), trained on significantly larger datasets. Additionally, for controlled comparisons, we also train our ViT-based tokenizer without denoising objective as a baseline.

**Our tokenizer implementation.** We implement our tokenizer using ViTs for both encoder and decoder (Vaswani et al., 2017; Dosovitskiy et al., 2021). We adopt recent architectural advances from LLaMA (Touvron et al., 2023), including RoPE (Su et al., 2024) (together with learned positional embeddings following Fang et al. (2023)), RMSNorm (Zhang & Sennrich, 2019), and SwiGLU-FFN (Shazeer, 2020). We use a patch size of 16, and the latent dimension is set to 16.

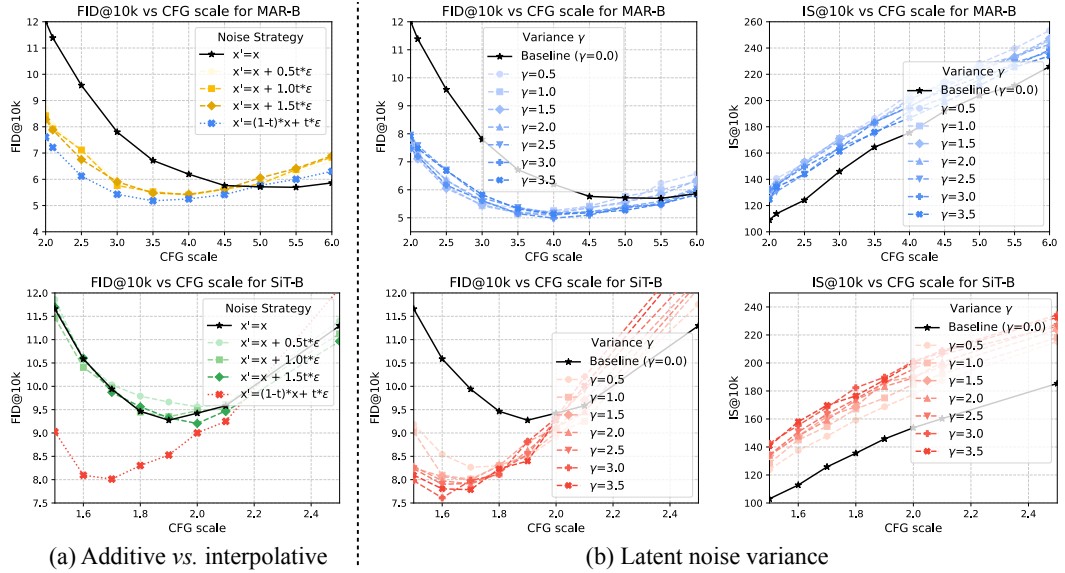

Figure 2: **Ablation on latent noise design. (a) Additive vs. interpolative noise.** Interpolative noise clearly outperforms additive noise for both MAR (Li et al., 2024a) and SiT (Ma et al., 2024). Both interpolative and additive latent noise lead to improved performance for MAR. **(b) Latent noise standard deviation ($\gamma$).** Our $l$-DeTok remains robust across various noise standard deviations. Generally, increasing $\gamma$ improves generation quality, with best results achieved around $\gamma = 3.0$.

**Tokenizer training.** We set the default weights in Eq. 6 to $\lambda_{\mathrm{KL}} = 10^{-6}$, $\lambda_{\mathrm{percep}} = 1.0$, and $\lambda_{\mathrm{GAN}} = 0.1$. In ablation studies, we use ViT-S for the encoder and ViT-B for the decoder, disable the GAN loss, and train for 50 epochs. For final experiments, we use ViT-B for both encoder and decoder, train for 200 epochs, and activate the GAN loss starting from epoch 100.

**Generative models.** To evaluate the broad effectiveness of a tokenizer, we experiment with *six* representative generative models, including three non-autoregressive models: DiT (Peebles & Xie, 2023), SiT (Ma et al., 2024), and LightningDiT (Yao & Wang, 2025); and three autoregressive models: MAR (Li et al., 2024a), causal RandomAR, and RasterAR based on RAR (Yu et al., 2024) and *diffloss* (Li et al., 2024a). We follow the officially released implementations to reimplement all methods within a unified codebase, standardizing training and evaluation across methods. Previous works on visual tokenizers often exclusively evaluate on non-AR models (*e.g.*, DiT, SiT), yet we find improvements from non-AR models *do not* necessarily generalize to AR models (more on this later). Our experiments aim to provide useful data points toward a more universal tokenizer.

**Generative model training.** We use a standardized training recipe for all generative models. Specifically, we follow the hyperparameters from Yao & Wang (2025), training generative models with a global batch size of 1024, using AdamW (Loshchilov & Hutter, 2019) with a constant learning rate of $2 \times 10^{-4}$, without warm-up, gradient clipping, or weight decay. Autoregressive models adopt the three-layer 1024-channel *diffloss* MLP from Li et al. (2024a). For ablation studies, we train generative models for 100 epochs. For larger-scale experiments on MAR models, we train for 800 epochs.

## 5 EXPERIMENTS

### 5.1 MAIN PROPERTIES

We use SiT-B (Ma et al., 2024) and MAR-B (Li et al., 2024a) as representative non-autoregressive (non-AR) and autoregressive (AR) generative models to study the generalizability of a tokenizer. Quantitative reconstruction results for tokenizers trained under different noises are in Sec. B.1.

### 5.1.1 PROPERTIES OF LATENT NOISING

We first study latent noising as the sole form of deconstruction, without any masking.

**Interpolative vs. additive noise.** An important design choice in our $l$-DeTok is the use of interpolative latent noise instead of additive noise, which we ablate here. Specifically, we compare two

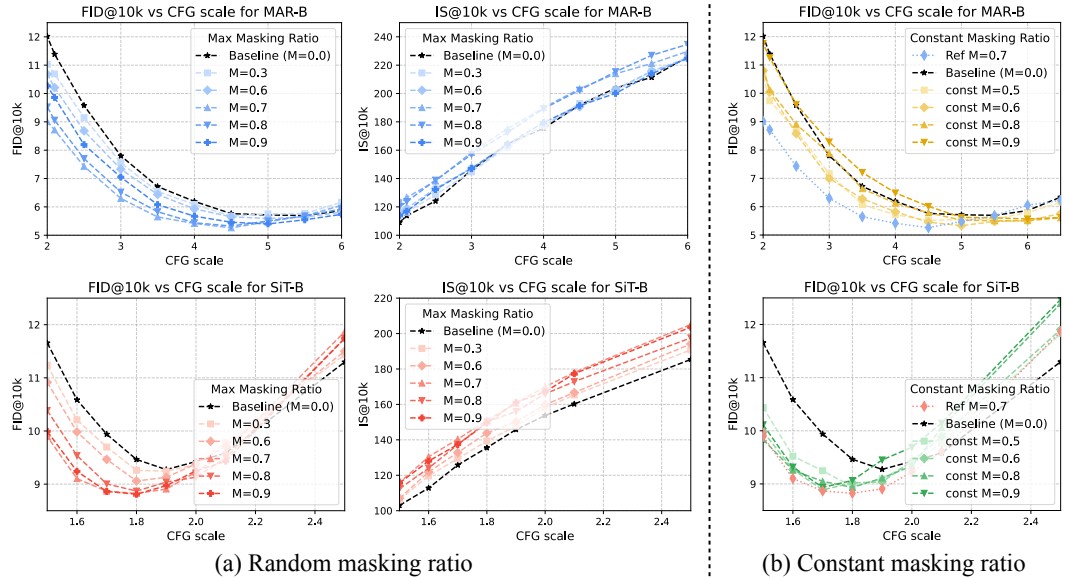

(a) Random masking ratio          (b) Constant masking ratio

Figure 3: **Ablation on masking ratio.** We show generative performance (FID↓) with varying masking ratios for tokenizers trained with **(a)** random and **(b)** constant masking ratio. Both MAR (Li et al., 2024b) (top) and SiT (Ma et al., 2024) (bottom) benefit from masking-based tokenizers, favoring heavy masking (70% to 90%). Randomized masking consistently outperforms constant masking.

latent noising variants: interpolative noise, *i.e.*, $\mathbf{x}' = (1 - \tau)\mathbf{x} + \tau\boldsymbol{\varepsilon}$ (Eq. 4), and additive noise, *i.e.*, $\mathbf{x}' = \mathbf{x} + \tau\boldsymbol{\varepsilon}$. We set the noise standard deviation to $\gamma = 1.0$ here. Figure 2-(a) presents the results: Interpolative noise clearly outperforms additive noise for both SiT (Ma et al., 2024) and MAR (Li et al., 2024a). This aligns with our expectation: interpolative noise ensures latent embeddings are heavily corrupted at high noise levels. In contrast, additive noise can potentially create shortcuts by making original signals remain dominant, reducing the effectiveness of the additive noise. Nonetheless, we observe the additive latent noise still improves MAR but not SiT.

**Noise standard deviation.** Figure 2-(b) studies the effect of noise standard deviation ($\gamma$ in Eq. 4). Both SiT and MAR consistently improve with interpolative latent noise across all tested levels. Performance peaks at moderately high standard deviation, indicating that stronger corruption generally yields more effective latents. This result confirms our key hypothesis: challenging *denoising* tasks naturally produce robust, downstream-aligned latents that benefit generative modeling.

**Noise schedule.** Inspired by Hoogeboom et al. (2023); Esser et al. (2024), we study how the sampling distribution of the noise level $\tau$ in Eq. 4 affects generation quality. Following Esser et al. (2024), we draw $\tau$ from a logit-normal distribution with mean $\mu$ (and fix $\gamma = 3.0$). As shown in Table 1, all uniform and non-uniform schedules improve FID over the $\tau = 0$ baseline, and concentrating more probability mass on higher noise levels ($\mu = 0.8$) yields the relatively best results for both MAR-B and SiT-B. This indicates that *l*-DeTok is robust to the choice of noise schedule and can still benefit from further tuning of the noise distribution.

| Sampling of $\tau$ | FID↓ | |
|---|---|---|
| | MAR-B | SiT-B |
| $\tau = 0$ (baseline) | 3.31 | 6.97 |
| $\tau \sim \mathcal{U}(0, 1)$ (default) | 2.77 | 5.56 |
| $\text{logit}(\tau) \sim \mathcal{N}(-0.8, 1)$ | 2.79 | 6.04 |
| $\text{logit}(\tau) \sim \mathcal{N}(0, 1)$ | 2.77 | 5.93 |
| $\text{logit}(\tau) \sim \mathcal{N}(0.8, 1)$ | **2.58** | **5.44** |

Table 1: **Impact of noise level sampling.**

### 5.1.2 PROPERTIES OF MASKING

We next investigate masking noise independently, without any latent noise.

**Masking ratio.** We examine how varying the maximal masking ratio $M$ (Eq. 5) influences generation quality. As shown in Figure 3-(a), both SiT and MAR benefit from masking-based tokenizer training in generation quality. Masking ratios between 70% and 90% consistently yield stronger performance compared to low masking ratios (*e.g.*, 30%), favoring high degrees of masking. This

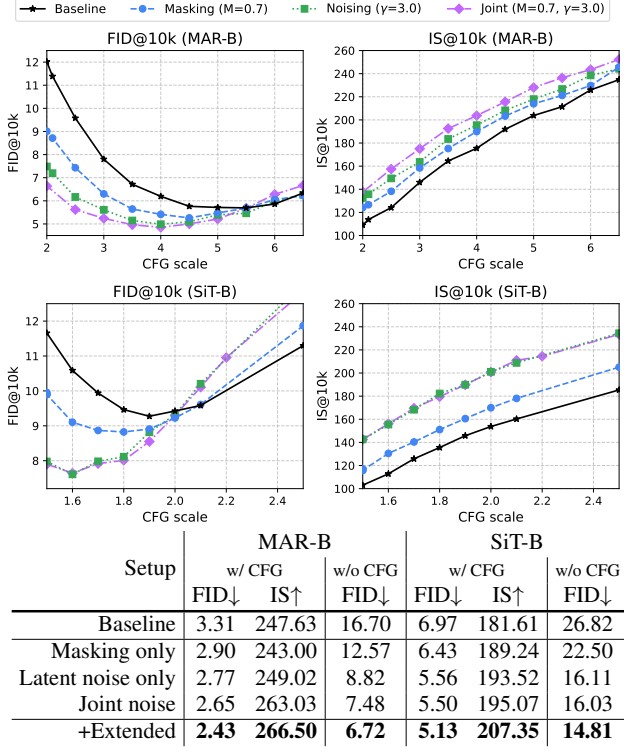

Figure 4: **Impact of tokenizer training strategies on generative performance.** We compare our baseline tokenizer with our $l$-DeTok variants: masking-only ($M=0.7$), latent noising-only ($\gamma=3.0$), and their combination (joint noising). Both masking and latent noising independently improve generation quality, with latent noising showing a stronger effect. Joint noising further improves performance for MAR, particularly in inception scores (IS), but provides limited additional benefit for SiT when latent noising is already applied. FID@50k scores are detailed in Tab. 2.

|  | MAR-B | | | SiT-B | | |
|---|---|---|---|---|---|---|
| Setup | w/ CFG | | w/o CFG | w/ CFG | | w/o CFG |
|  | FID↓ | IS↑ | FID↓ | FID↓ | IS↑ | FID↓ |
| Baseline | 3.31 | 247.63 | 16.70 | 6.97 | 181.61 | 26.82 |
| Masking only | 2.90 | 243.00 | 12.57 | 6.43 | 189.24 | 22.50 |
| Latent noise only | 2.77 | 249.02 | 8.82 | 5.56 | 193.52 | 16.11 |
| Joint noise | 2.65 | 263.03 | 7.48 | 5.50 | 195.07 | 16.03 |
| +Extended | **2.43** | **266.50** | **6.72** | **5.13** | **207.35** | **14.81** |

Table 2: **Effectiveness of denoising**. We report FID and IS evaluated on 50,000 images here. Compared to baselines, we see substantial gains in generative models when using our $l$-DeTok. Extended: larger encoder, longer training, GAN enabled midway through training.

behavior mirrors observations from latent denoising, *i.e.*, challenging denoising is more beneficial, indicating a common underlying principle under the *deconstruction-reconstruction* strategy.

**Constant *vs.* randomized masking ratio.** Figure 3-(b) compares randomized masking ratios against constant ones. For constant ratios, we further fine-tune the tokenizer decoder for 10 epochs using *fully-visible* latents to alleviate the distribution shift between training and inference since fully-visible inputs are absent in constant-ratio training. This adjustment improves rPSNR by 1–3 and rFID by 0.6 (details in Sec. B.1). Figure 3-(b) shows randomized masking consistently outperforms constant masking, as it encourages latent embeddings to be robust to varying corruption levels. This aligns naturally with downstream tasks that require denoising across diverse corruption levels.

### 5.1.3 JOINT DENOISING

Prior ablations indicate that both latent noising and masking can independently improve generation quality, with latent noising showing a stronger effect. Here, we investigate the effect of joint denoising. Based on prior results, we fix the noise standard deviation to $\gamma = 3.0$ and the masking ratio to $M = 0.7$. Figure 4 and Table 2 summarize the results. With joint denoising, our $l$-DeTok achieves FID scores of 5.50 (SiT-B) and 2.65 (MAR-B) with CFG (Table 2). In comparison, our baseline tokenizer—trained with identical settings but without any noise—obtains worse results: 6.97 (SiT-B) and 3.31 (MAR-B). We observe that joint denoising is more effective for MAR, but provides limited additional benefit for SiT when latent noising is already applied. This indicates that latent denoising is essential, while the masking-based denoising is *optional*.

Lastly, with joint denoising, we increase the encoder to base size, train for 200 epochs, and enable the GAN loss starting from epoch 100 (+Extended in Table 2). Under this setting, FID improves to 5.13 (SiT) and 2.43 (MAR). We adopt this improved tokenizer for all subsequent evaluations.

### 5.1.4 SCALABILITY

We study whether the gains from $l$-DeTok persist under generative model scaling. We train SiT-B/L/XL and MAR-B/L for 100 epochs, comparing the baseline tokenizer and ours under identical settings. As shown in Table 3, $l$-DeTok consistently improves FID at all model sizes for both SiT and MAR. The advantage remains when we extend training to 800 epochs in the system-level comparison of Table 5.

| Model | FID↓ (w/ CFG) | |
|---|---|---|
|  | Baseline | **Ours** |
| SiT-B | 7.08 | **5.13** |
| SiT-L | 4.66 | **3.49** |
| SiT-XL | 4.47 | **3.14** |
| MAR-B | 3.65 | **2.43** |
| MAR-L | 2.44 | **2.08** |

Table 3: **Scalability.**

Table 4: **Generalizability comparison of tokenizers across different generative models.** We compare various tokenizers on representative generative models. Our *l*-DeTok tokenizer outperforms other tokenizers for AR models, and also surpasses standard tokenizers trained without semantics distillation for non-AR models. All results are obtained with optimal CFG scales.

| Tokenizer | rFID↓ | Autoregressive Models | | | | | | Non-autoregressive Models | | | | | |
| | | MAR | | RandomAR | | RasterAR | | SiT | | DiT | | Light.DiT | |
| | | FID↓ | IS↑ | FID↓ | IS↑ | FID↓ | IS↑ | FID↓ | IS↑ | FID↓ | IS↑ | FID↓ | IS↑ |
| *Tokenizers trained with semantics distillation from external pretrained models* | | | | | | | | | | | | | |
| VA-VAE (Yao & Wang, 2025) | 0.28 | 16.66 | 144.5 | 38.13 | 68.3 | 15.88 | 160.5 | 4.33 | 221.1 | 4.91 | 213.9 | 2.86 | **275.1** |
| MAETok (Chen et al., 2025a) | 0.48 | 6.99 | 201.8 | 24.83 | 97.6 | 15.92 | 127.2 | 4.77 | **243.2** | 5.24 | **224.7** | 3.92 | 273.3 |
| Our *l*-DeTok + Distillation | 0.85 | 2.52 | 254.1 | 5.57 | 180.3 | 11.99 | 158.8 | **3.40** | 232.6 | **3.91** | 221.7 | **2.18** | 243.0 |
| *Tokenizers trained without semantics distillation* | | | | | | | | | | | | | |
| SD-VAE (Rombach et al., 2022) | 0.61 | 4.64 | 259.8 | 13.11 | 141.8 | 8.26 | 179.3 | 7.66 | 187.5 | 8.33 | 179.8 | 4.24 | 223.7 |
| SD3-VAE (Esser et al., 2024) | **0.21** | 6.46 | 250.0 | 40.65 | 70.89 | 21.03 | 101.7 | 9.57 | 170.8 | 13.63 | 126.8 | 5.52 | 222.2 |
| MAR-VAE (Li et al., 2024a) | 0.53 | 3.71 | 265.3 | 11.78 | 147.9 | 7.99 | 189.7 | 6.26 | 177.5 | 8.20 | 171.8 | 3.98 | 218.7 |
| Our *l*-DeTok | 0.68 | **2.43** | **266.5** | **5.22** | **248.9** | **4.46** | **257.7** | 5.13 | 207.3 | 6.58 | 173.9 | 3.63 | 225.4 |

## 5.2 GENERALIZATION EXPERIMENTS

To comprehensively evaluate tokenizer generalizability, we compare performance across six representative generative models: three non-autoregressive (DiT (Peebles & Xie, 2023), SiT (Ma et al., 2024), LightningDiT (Yao & Wang, 2025)) and three autoregressive (MAR (Li et al., 2024a), RandomAR, RasterAR). RandomAR and RasterAR are adapted from RAR (Yu et al., 2024). We modify them to support decoding continuous tokens via *diffloss* MLPs (Li et al., 2024a). RandomAR generates tokens in a random order while RasterAR uses a raster-scan order. We use base-sized models and train them for 100 epochs for experiments here.

**Comparisons with standard convolutional tokenizers.** Our *l*-DeTok tokenizer consistently outperforms conventional tokenizers (Rombach et al., 2022; Esser et al., 2021; 2024), by large margins across both AR and non-AR models. Table 4 presents the results. Compared to the best existing tokenizer (MAR-VAE), our method significantly improves FID (with CFG) from 3.71 to 2.43 (∼34%) for MAR, from 11.78 to 5.22 (∼56%) for RandomAR, and from 7.99 to 4.46 (∼44%) for RasterAR. Improvements for non-autoregressive models are also consistent.

**Comparisons with semantics-distilled tokenizers.** Our *l*-DeTok generalizes significantly better than prior semantics-distilled tokenizers. Table 4 compares our method with recent approaches such as VA-VAE (Yao & Wang, 2025) and MAETok (Chen et al., 2025a), which distill semantics from pretrained encoders. *Surprisingly*, we empirically find these tokenizers, despite promising performance in non-AR models, do not generalize well to AR models. Previous studies implicitly assume that tokenizer improvements from one generative paradigm naturally transfer to others. However, our experiments *challenge* this assumption, revealing a previously unrecognized gap: tokenizer effectiveness in one paradigm *does not* necessarily transfer to others.

In sharp contrast, our method generalizes robustly across both non-AR and AR models. Critically, our *l*-DeTok achieves this *without* any semantics distillation from pretrained encoders. To further investigate *whether* our *l*-DeTok can benefit from semantics distillation, we incorporate an auxiliary semantics-distillation loss (details in Sec. A.4). Remarkably, this *privileged* version of our tokenizer achieves the best FID scores for non-AR models, surpassing *all* previous semantics-distilled tokenizers. Although AR models benefit relatively less from distillation and may even slightly degrade, their performance still remains substantially superior to previous methods.

***l*-DeTok with convolutional tokenizers.** Our *l*-DeTok generalizes across Transformer- and convolution-based tokenizers, as the idea of *latent denoising* is architecture-agnostic. To demonstrate this, we train CNN-based tokenizers for 50 epochs with and without our denoising objective (details in Sec. A.5). With the default CNN baseline, MAR-B achieves 3.32 FID and SiT-B 7.11 FID at best CFGs. With our denoising-based CNN tokenizers, MAR-B improves to 2.82 FID and SiT-B to 5.62 FID, demonstrating that *l*-DeTok provides consistent gains also in convolutional networks.

## 5.3 BENCHMARKING WITH PREVIOUS SYSTEMS

We compare against leading generative systems in Table 5. For this experiment, we train MAR-B and MAR-L for 800 epochs. Simply adopting our tokenizer substantially improves the generative performance: MAR-B achieves an FID of 1.55 (from 2.31), and MAR-L further improves to 1.35 (from 1.78). Notably, our MAR-B and MAR-L both match or surpass the previously best-performing

Table 5: **System-level comparison** on ImageNet 256×256 class-conditioned generation. Our approach enables MAR models (Li et al., 2024a) to achieve leading results without relying on semantics distillation. †: With additional decoder fine-tuning (see Sec. A.3 for details).

| | | w/o CFG | | | | w/ CFG | | | |
|---|---|---|---|---|---|---|---|---|---|
| | #params | FID↓ | IS↑ | Pre.↑ | Rec.↑ | FID↓ | IS↑ | Pre.↑ | Rec.↑ |
| *With semantics distillation from external pretrained models* | | | | | | | | | |
| SiT-XL + REPA (Yu et al., 2025b) | 675M | 5.90 | 157.8 | 0.70 | 0.69 | 1.42 | 305.7 | 0.80 | 0.64 |
| SiT-XL + MAETok (Chen et al., 2025a) | 675M | 2.31 | **216.5** | - | - | 1.67 | **311.2** | - | - |
| LightningDiT + MAETok (Chen et al., 2025a) | 675M | 2.21 | 208.3 | - | - | 1.73 | 308.4 | - | - |
| LightningDiT + VAVAE (Yao & Wang, 2025) | 675M | **2.17** | 205.6 | 0.77 | 0.65 | 1.35 | 295.3 | 0.79 | 0.65 |
| DDT-XL (Wang et al., 2025) | 675M | 6.27 | 154.7 | 0.68 | 0.69 | **1.26** | 310.6 | 0.79 | 0.65 |
| *Without semantics distillation from external pretrained models* | | | | | | | | | |
| DiT-XL/2 (Peebles & Xie, 2023) | 675M | 9.62 | 121.5 | 0.67 | 0.67 | 2.27 | 278.2 | 0.83 | 0.57 |
| SiT-XL/2 (Ma et al., 2024) | 675M | 8.30 | - | - | - | 2.06 | 270.3 | 0.82 | 0.59 |
| VAR-d30 (Tian et al., 2025) | 2.0B | - | - | - | - | 1.92 | **323.1** | 0.82 | 0.59 |
| LlamaGen-3B (Sun et al., 2024) | 3.1B | - | - | - | - | 2.18 | 263.3 | 0.81 | 0.58 |
| RandAR-XXL (Pang et al., 2025) | 1.4B | - | - | - | - | 2.15 | 322.0 | 0.79 | 0.62 |
| CausalFusion (Deng et al., 2024) | 676M | 3.61 | 180.9 | 0.75 | 0.66 | 1.77 | 282.3 | 0.82 | 0.61 |
| MAR-B + MAR-VAE (Li et al., 2024a) | 208M | 3.48 | 192.4 | 0.78 | 0.58 | 2.31 | 281.7 | 0.82 | 0.57 |
| MAR-L + MAR-VAE (Li et al., 2024a) | 479M | 2.60 | 221.4 | 0.79 | 0.60 | 1.78 | 296.0 | 0.81 | 0.60 |
| MAR-H + MAR-VAE (Li et al., 2024a) | 943M | 2.35 | 227.8 | 0.79 | 0.62 | 1.55 | 303.7 | 0.81 | 0.62 |
| MAR-B (Li et al., 2024a) + our *l*-DeTok | 208M | 2.79 | 195.9 | 0.80 | 0.60 | 1.61 | 289.7 | 0.81 | 0.62 |
| MAR-B (Li et al., 2024a) + our *l*-DeTok† | 208M | 2.94 | 195.5 | 0.80 | 0.59 | 1.55 | 291.0 | 0.81 | 0.62 |
| MAR-L (Li et al., 2024a) + our *l*-DeTok | 479M | **1.84** | 238.4 | 0.82 | 0.60 | 1.43 | 303.5 | 0.82 | 0.61 |
| MAR-L (Li et al., 2024a) + our *l*-DeTok† | 479M | 1.86 | **238.6** | 0.82 | 0.61 | **1.35** | 304.1 | 0.81 | 0.62 |

| Model | #params | FID↓ | IS↑ |
|---|---|---|---|
| ADM (Dhariwal & Nichol, 2021) | 554M | 7.72 | 172.7 |
| DiT-XL/2 (Peebles & Xie, 2023) | 675M | 3.04 | 240.8 |
| SiT-XL/2 (Ma et al., 2024) | 675M | 2.62 | 252.2 |
| SiT-XL/2 + REPA (Yu et al., 2025b) | 675M | 2.08 | 274.6 |
| MAR-L + MAR-VAE (Li et al., 2024a) | 479M | 1.73 | 279.9 |
| MAR-B + Ours (scratch, 400ep) | 208M | 1.83 | 279.6 |
| MAR-L + Ours (fine-tune, 200ep) | 479M | **1.61** | 315.7 |

Table 6: **System-level comparison** on ImageNet 512×512 class-conditioned generation. Our tokenizer is fine-tuned from ImageNet 256×256 checkpoint for 25 epochs. Scratch *vs.* fine-tune: Generative models trained from scratch or initialized from ImageNet 256×256 checkpoints (see Sec. A.3 for details).

huge-size MAR model (1.35 *vs.* 1.55). Qualitative results are provided in Figure 5. We also report results on ImageNet $512 \times 512$ in Table 6. Our MAR-B variant already matches MAR-L, a 2.3× larger model, when trained from scratch. Our MAR-L achieves remarkable 1.61 FID and 315.7 IS.

## 5.4 TEXT-TO-IMAGE GENERATION

We further validate *l*-DeTok on text-to-image (T2I) generation. Following the protocol in Bao et al. (2022); Yu et al. (2025b), we train models from scratch on the MS-COCO train split (Lin et al., 2014) and evaluate on the validation split using FID-30k. We compare T2I variants of MAR-B and SiT-B under identical conditions, varying only the tokenizer (details in Sec. C). Table 7 summarizes the results, demonstrating that *l*-DeTok substantially improves not only sample quality and diversity (FID), but also text–image conditioning alignment (CLIP score). Qualitatively (see Figures. C.1, C.2,C.3, and C.4), other tokenizers frequently produce the "spot artifacts" reported in previous works (Fan et al., 2025; Team et al., 2025) under the text-to-image setting, whereas such artifacts are notably absent with *l*-DeTok.

| Tokenizer | T2I MAR-B | | T2I SiT-B | |
|---|---|---|---|---|
| | FID↓ | CLIP↑ | FID↓ | CLIP↑ |
| VA-VAE (Yao & Wang, 2025) | 34.64 | 21.98 | 5.83 | **25.07** |
| SD-VAE Rombach et al. (2022) | 19.75 | 22.86 | 6.63 | 24.00 |
| MAR-VAE (Li et al., 2024a) | 12.49 | 22.07 | 5.74 | 23.50 |
| **Ours (*l*-DeTok)** | **4.97** | **24.82** | **4.31** | 24.61 |

Table 7: **MS-COCO text-to-image generation.** Comparison of tokenizers on T2I-variants MAR-B and SiT-B, reported with best CFG scales. Our *l*-DeTok achieves both lower FID (better diversity) and higher CLIP scores (better alignment), outperforming all others.

## 5.5 BEYOND STANDARD CONTINUOUS TOKENIZERS

To assess the generality of our approach beyond standard 2D continuous tokenizers, we study both 1D continuous tokenizers and discrete vector-qunatized (VQ) tokenizers.

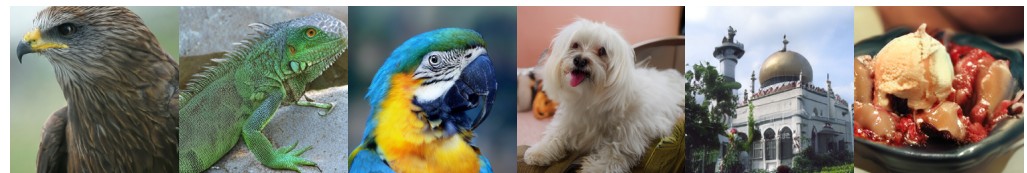

Figure 5: **Qualitative Results.** We show selected examples of class-conditional generation on ImageNet 256×256 using MAR-L (Li et al., 2024a) trained with our tokenizer.

**1D Tokenizers.** We first test $l$-DeTok with recent 1D tokenizers that map images to compact 1D token sequences (Yu et al., 2025a; Yan et al., 2025; Duggal et al., 2025; Bachmann et al., 2025). These tokenizers substantially reduce the sequence length compared to standard 2D latent grids. Sec. A.6 provides implementation details and an interesting pitfall coupled with the "grokking" phenomenon we observe in these 1D tokenizers. As shown in Table 8, $l$-DeTok improves generation performance over the corresponding baselines, with gains becoming more pronounced at higher token budgets. This suggests that the benefit of *denoising* is not tied to a particular latent topology, and remains effective even when the representation is compressed and less-structured.

**Vector-quantized Tokenizers.** We next evaluate $l$-DeTok on VQ tokenizers, where the latents are discrete code indices instead of continuous features. In this setting, the idea of removing *masking* noise applies directly. Section A.7 presents implementation details. We consider both random-order (RandomAR) and standard raster-scan auto-regressive models (RasterAR). As summarized in Table 9, $l$-DeTok improves both variants: for RandomAR, FID is improved from 8.48 to 7.03; for RasterAR, FID improves from 5.29 to 4.91, with IS also increasing in both cases. These gains indicate the broader applicability of our *denoising* principle beyond continuous tokenizers.

Table 8: **1D continuous tokenizers on ImageNet 256×256.** Comparison of MAR and SiT with different numbers of 1D tokens, with and without our denoising training, evaluated by FID@50k.

| #Tokens | Setting | MAR-B | | SiT-B | |
|---|---|---|---|---|---|
| | | w/ CFG | w/o CFG | w/ CFG | w/o CFG |
| 32 | Baseline | **5.22** | 17.63 | **6.99** | 18.91 |
| | **+Ours** | 5.51 | **14.87** | 7.02 | **16.46** |
| 64 | Baseline | 4.58 | 17.73 | 7.05 | 22.82 |
| | **+Ours** | **4.12** | **14.02** | **6.40** | **18.59** |
| 128 | Baseline | 3.92 | 18.63 | 7.95 | 28.36 |
| | **+Ours** | **3.14** | **12.05** | **6.20** | **20.22** |

Table 9: **Vector-quantized tokenizers on ImageNet 256×256.** Comparison of VQ tokenizers with and without our denoising training in RandomAR and RasterAR models, evaluated by FID@50k and IS@50k with CFGs. $l$-DeTok improves generative quality across generation orders.

| Setting | RandomAR-B | | RasterAR-B | |
|---|---|---|---|---|
| | FID↓ | IS↑ | FID↓ | IS↑ |
| Baseline VQ | 8.48 | 144.93 | 5.29 | 204.15 |
| **+Ours** | **7.03** | **162.68** | **4.91** | **208.61** |

## 6 DISCUSSION AND CONCLUSION

Simple principles that scale well are at the core of generative modeling. In this work, we have shown that a *surprisingly simple* principle, *i.e.*, *denoising*-based tokenizer, already provides consistent benefits across a broad design space, including non-autoregressive and autoregressive generative models, 2D/1D/VQ tokenizers, CNN and Transformer-based architectures, and both raster and random generation orders, while adding almost no system complexity. Several questions remain open.

*First*, our experiments are conducted within a moderate-scale regime ($\mathcal{O}(10^6)$ samples), where $l$-DeTok improves performance when we scale up within this envelope. A natural next step is to test it in the frontier regime of web-scale models and datasets ($\mathcal{O}(10^7)$–$\mathcal{O}(10^{10})$ samples), more challenging settings such as video generation, and to understand how denoising-aligned tokenizers behave in those cases. *Second*, our results also parallel the recent trend of semantics distillation from pre-trained encoders such as DINOv2. Distillation is effective when strong teachers exist and align well with the target domain, but it *may* become a bottleneck as we move to larger data and broader distributions that *surpass* the teacher's coverage, or to domains without reliable teachers. In contrast, $l$-DeTok is self-contained and does not depend on external teachers. At scale, we hope such task-aligned tokenizers *may* provide a more flexible path forward. *Finally*, reconstructing clean inputs from noisy latents makes our tokenizer objective reminiscent of $x_0$-prediction in modern generative models. Understanding the boundary—and eventual unification—among *reconstruction*, *denoising* and *generation* is an intriguing direction, *e.g.*, Li & He (2025). We hope this perspective will help guide future work on scalable generative models and the tokenizers that support them.

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

APPENDIX

# A TRAINING AND INFERENCE DETAILS

## A.1 TOKENIZER

**Model.** We implement our tokenizer using ViTs for both encoder and decoder (Vaswani et al., 2017; Dosovitskiy et al., 2021). We provide detailed model parameters in Table A.1. We adopt recent architectural advances from LLaMA (Touvron et al., 2023), including RoPE (Su et al., 2024) (together with learnable positional embeddings following Fang et al. (2023)), RMSNorm (Zhang & Sennrich, 2019), and SwiGLU-FFN (Shazeer, 2020). The encoder operates at a patch size of 16, yielding 256 latent tokens for each $256 \times 256$ image, while the decoder uses patch size 1 as there is no resolution change. The latent dimension is set to 16. We omit the `[CLS]` token for simplicity.

Table A.1: **Model Size.** We report the model configurations and parameter counts of tokenizer encoders and decoders. The total tokenizer size is the sum of encoder and decoder parameters. For example, a tokenizer combining a ViT-S encoder and ViT-B decoder (S-B) has 111.6M parameters, while a tokenizer with both ViT-B encoder and decoder (B-B) has 171.7M parameters.

| Size | Hidden Size | Blocks | Heads | Parameters |
|---|---|---|---|---|
| Small (S) | 512 | 8 | 8 | 25.75M |
| Base (B) | 768 | 12 | 12 | 85.85M |
| Large (L) | 1024 | 24 | 16 | 303.33M |

**Training.** Our tokenizer is trained using the weighted loss defined in Eq. 6, with default weights $\lambda_{\text{KL}} = 10^{-6}$, $\lambda_{\text{percep}} = 1.0$, and $\lambda_{\text{GAN}} = 0.1$. For ablation studies, we use a ViT-S encoder and a ViT-B decoder, disable GAN loss, and train for 50 epochs. We observe that including a GAN loss (Yu et al., 2021; Goodfellow et al., 2014; Esser et al., 2021) sharpens reconstructions but roughly doubles training time, without altering result trends. For our final experiments, we adopt ViT-B for both encoder and decoder, enable GAN loss from epoch 100, and train for 200 epochs. In both settings, we use a global batch size of 1024, and the peak learning rate is set to $4.0 \times 10^{-4}$ (scaled linearly from $1.0 \times 10^{-4}$ at a batch size of 256 (Goyal et al., 2017)). We apply linear warm-up for 25% of the total epochs (12 epochs for 50-epoch training, and 50 epochs for 200-epoch training), followed by cosine learning rate decay. We use the AdamW optimizer (Loshchilov & Hutter, 2019) with $\beta$ parameters (0.9, 0.95) and a weight decay of $1.0 \times 10^{-4}$. The only data augmentation employed is horizontal flipping. Our reconstruction loss closely follows the implementation in Yu et al. (2025a).

**Training budget.** Training an S-B tokenizer (see Table A.1) without GAN loss for 50 epochs takes roughly 160 NVIDIA A100 GPU hours (enabling GAN loss from epoch 20 extends training to about 288 A100 GPU hours). Training a B-B tokenizer with GAN loss enabled from epoch 100 for a total of 200 epochs takes approximately 1,150 A100 GPU hours.

We present a detailed comparison of training cost between our baseline and $l$-DeTok tokenizers in Table A.2. All configurations use the same architecture, batch size (1024), and total number of epochs; the only difference is whether we use the baseline objective or $l$-DeTok. As shown, the approximate A100 hours of our method are essentially identical to, or slightly lower than, the baseline: S-B goes from 180 (baseline) to 160–180 hours with $l$-DeTok, B-B from 1150 to 1140 hours, and the 1D and VQ variants remain within the same 170–190 hour range. This indicates that the denoising mechanism does not introduce any meaningful extra computational overhead beyond the standard tokenizer training; in practice, the variation is within normal run-to-run fluctuation.

**Pseudo-code of latent denoising**. See Algorithm 1.

## A.2 GENERATIVE MODELS

**Model.** To evaluate the broad effectiveness of a tokenizer, we experiment with *six* representative generative models, including three non-autoregressive models: DiT (Peebles & Xie, 2023), SiT (Ma et al., 2024), and LightningDiT (Yao & Wang, 2025); and three autoregressive models: MAR (Li et al., 2024a), causal RandomAR, and RasterAR based on RAR (Yu et al., 2024) and *diffloss* (Li

Table A.2: **Training Cost.** Comparison of training schedules and approximate compute for different tokenizer and setting combinations. Our models are trained across mixed compute and storage environments (*e.g.*, A100s, H100s, and A6000s). We report approximate A100-equivalent hours over the full training run. Size: Encoder size - Decoder size. S: Small, B: Base.

| Size | Setting | Epochs | Batch size | GAN start epoch | Approx. A100 Hours |
|------|---------|--------|-----------|-----------------|---------------------|
| S-B | Baseline | 50 | 1024 | disabled | 180 |
| S-B | *l*-DeTok | 50 | 1024 | disabled | 160–180 |
| B-B | Baseline | 200 | 1024 | 100 | 1150 |
| B-B | *l*-DeTok | 200 | 1024 | 100 | 1140 |
| B-B | *l*-DeTok + Distill. | 200 | 1024 | 100 | 1900 |
| S-B | 1D-baseline | 50 | 1024 | disabled | 170–190 |
| S-B | 1D-*l*-DeTok | 50 | 1024 | disabled | 170–190 |
| S-B | VQ-baseline | 50 | 1024 | disabled | 190 |
| S-B | VQ-*l*-DeTok | 50 | 1024 | disabled | 170 |

---

**Algorithm 1** Latent Denoising: PyTorch-like Pseudo-code

---

```python
def denoise(x, encoder, decoder, max_mask_ratio=0.7, gamma=3.0):
    # encode input image to latent embeddings under (optional) masking
    z, ids_restore = encoder(x, max_mask_ratio=max_mask_ratio)

    # variational latent embeddings
    posteriors = diagonal_gaussian_dist(z)
    z_sampled = posteriors.sample()

    # sample interpolation factor uniformly from [0, 1]
    bsz, n_tokens, chans = z_sampled.shape
    device = z_sampled.device
    noise_level = torch.rand(bsz, 1, 1, device=device).expand(-1, n_tokens,
        chans)

    # generate Gaussian noise
    noise = gamma * torch.randn(bsz, n_tokens, chans, device=device)

    # interpolate latent embeddings with noise
    z_noised = (1 - noise_level) * z_quantized + noise_level * noise

    # reconstruct the inputs
    recon = decoder(z_noised, ids_restore)
    return recon
```

---

et al., 2024a). We follow their officially released code to re-implement all generative models within our codebase to standardize training and evaluation, ensuring fair comparisons.

We introduce some modifications: for DiT (Peebles & Xie, 2023), SiT (Ma et al., 2024), and LightningDiT (Yao & Wang, 2025), we apply classifier-free guidance (CFG) to all latent channels, rather than only the first three channels used in their original implementations.[2] For training on 1D tokens (*e.g.*, MAETok (Chen et al., 2025a)), we use simple 1D learnable positional embeddings and disable RoPE used in LightningDiT. We adopt the default samplers from the original implementations, using 250 denoising steps during inference. Due to the frequently observed training instability in SiT and LightningDiT in our early experiments, we apply QK-Norm (Dehghani et al., 2023) to stabilize training. These modifications are consistent across all tokenizers for fairness.

For autoregressive (AR) methods, we use the default MAR model and implement RandomAR and RasterAR following RAR (Yu et al., 2024). To produce continuous tokens, we employ the *diffloss* (Li et al., 2024a) with a 3-layer, 1024-channel MLP head. We disable dropout in all MLP layers within Transformer blocks for autoregressive models. For inference, we use 64 autoregressive steps and 100 denoising steps for all experiments, except those in system-level comparison (Tables 5 and 6), where we use 256 autoregressive steps or 512 autoregressive steps. Sampling is performed with the default MAR sampler across all AR models. Following MAR (Li et al., 2024a), we set the sampling temperature to 1.0 when using CFG, and sweep temperatures when CFG is disabled (*i.e.*, CFG=1.0).

---

[2]See original implementations in DiT, SiT, and LightningDiT.

**Training.** We use a standardized training recipe for all generative models. Specifically, we follow the hyperparameters from Yao & Wang (2025), training generative models with a global batch size of 1024, using AdamW optimizer (Loshchilov & Hutter, 2019) with a constant learning rate of $2 \times 10^{-4}$, without warm-up, gradient clipping, or weight decay. We do not tune these hyperparameters. For ablation studies, we train generative models for 100 epochs. For larger-scale experiments on MAR models, we train them for 800 epochs. All models utilize exponential moving average (EMA) with a decay rate of 0.9999.

**Standardization.** We always standardize tokenizer outputs by subtracting the channel-wise mean and dividing by the channel-wise standard deviation, both computed from the ImageNet training set. For publicly available tokenizers, we use their official standardization steps.

**Metrics.** Unless noted otherwise, we report FID@50k scores with classifier-free guidance (CFG), with optimal CFG scales searched from FID@10k results. We abbreviate FID@50k as FID.

**Training budget.** Training DiT-B, SiT-B, and LightningDiT-B for 100 epochs takes approximately 128 to 200 A100 hours using locally cached tokens. Training MAR-B, RandomAR-B, and RasterAR-B for 100 epochs takes approximately 220 to 250 A100 hours. For the final MAR-B model used in the system-level comparison (Table 5), training for 800 epochs takes roughly 2,450 A100 hours; training MAR-L for the same duration takes about 3,850 A100 hours (online evaluation time included).

### A.3 TOKENIZER TUNING

**Decoder fine-tuning.** We observe an interesting discrepancy between training and inference in our $l$-DeTok. Our decoder is trained predominantly with noise-corrupted latent embeddings but encounters nearly clean embeddings during inference. To address this gap, we fine-tune the decoder on clean latent embeddings, *i.e.*, masking and latent noising are disabled, for an additional 100 epochs. This adjustment partially alleviates the discrepancy, improving the FID of the 800-epoch MAR-L model from 1.43 to 1.35 and MAR-B model from 1.61 to 1.55 (Table 5). Figure A.1 and Figure A.2 compare the denoising capability of different tokenizers. Although fine-tuning decoder reduces the decoder's denoising strength, it enhances the image quality (FID and IS) when decoding from generated latent embeddings. Crucially, this demonstrates that the quality of latent representations produced by the tokenizer *encoder*, rather than the *decoder*'s error-tolerance, is the primary driver of generative model improvements. Therefore, our $l$-DeTok does more than improving the the decoder's robustness to sampling errors.

To further verify that performance gains primarily stem from improved latent representations rather than enhanced decoder-side error tolerance, we conduct an additional experiment by fine-tuning only the MAR-VAE decoder on the denoising task while keeping the encoder frozen. In this way, we can reuse pre-trained models for comparison since their modeling space, *encoder latent space*, remains fixed. Contrary to expectations, this setup—improving decoder robustness without altering latent embeddings—actually leads to slightly worse FIDs (around 0.1 to 0.3 degradation). This indicates that merely boosting decoder denoising capability exacerbates the training-inference discrepancy, as generative models are already proficient at denoising. These results reinforce that the primary advantage of our approach lies explicitly in the improved quality of latent embeddings, not in decoder-side denoising enhancements.

**Adopting tokenizers to $512 \times 512$ resolution.** To save computation, we fine-tune tokenizer checkpoints from ImageNet-$256 \times 256$ for the ImageNet-$512 \times 512$ generation task. Specifically, we halve the learning rate to $2.0 \times 10^{-4}$ and fine-tune tokenizers initialized from the 200-epoch $256 \times 256$ checkpoints for an additional 25 epochs, enabling GAN loss from the start. We compare full fine-tuning to decoder-only fine-tuning: decoder-only fine-tuning achieves better PSNR (26.3 *vs.* 24.9) and comparable rFID (0.92 *vs.* 0.86), yet fully fine-tuned tokenizers consistently provide superior downstream generative performance. For instance, using fully fine-tuned tokenizers, MAR-B achieves 2.66 FID and SiT achieves 5.98 FID, compared to decoder-only fine-tuned tokenizers at 3.63 and 6.32 FID, respectively. All generative models are trained from scratch for 100 epochs, and we report FID@50K at optimal CFG scales. Consequently, we adopt fully fine-tuned tokenizers.

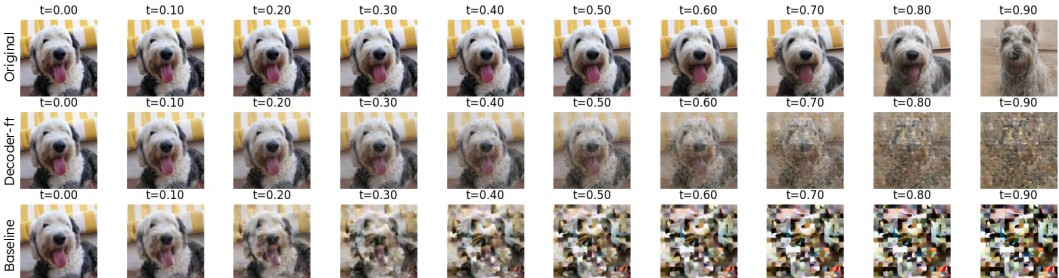

Figure A.1: **Visualization of latent denoising.** Images generated from latent embeddings corrupted with varying noise levels ($t$) by the original decoder (top), fine-tuned decoder (middle), and baseline decoder (bottom). The fine-tuned decoder shows a reduced ability to recover images from noisy embeddings compared to the original decoder. The baseline tokenizer trained without the denoising objective fails to reconstruct original images from noisy latents.

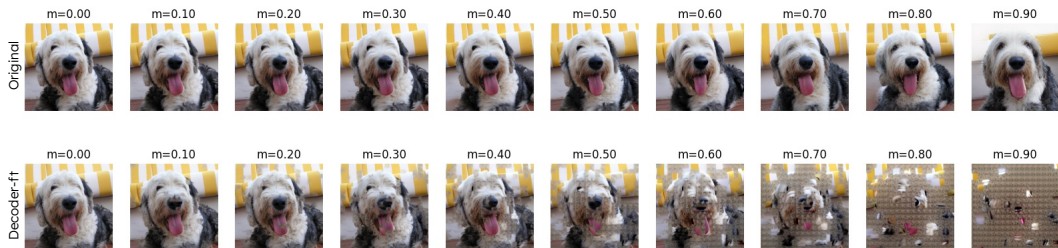

Figure A.2: **Visualization of "mask" denoising.** Images generated from masked inputs with varying masking ratios ($m$) by the original decoder (top) and fine-tuned decoder (bottom). The fine-tuned decoder exhibits diminished capability in reconstructing masked regions compared to the original decoder.

These results reinforce our core motivation: encoder-produced tokenizer embeddings primarily drive generative quality, while decoder fine-tuning provides incremental improvements. Therefore, improving encoder is more fundamental than improving the decoder.

**Adapting generative models to $512 \times 512$ resolution.** Similar to tokenizer fine-tuning, we fine-tune a MAR-L checkpoint from ImageNet-$256 \times 256$ for the ImageNet-$512 \times 512$ generation task to reduce compute. Additionally, we train a MAR-B model from scratch to verify the compatibility of our tokenizer for end-to-end training. Note that we do not aim to hill climb the performance metrics in Table 6, since our tokenizer is only fine-tuned for 25 epochs rather than trained from scratch for many epochs. Instead, our intention is for these results to demonstrate the broad applicability of our approach.

### A.4 $l$-DeTok with Semantics Distillation

Our $l$-DeTok *already* holds great promise for generative models across both autoregressive (AR) and non-autoregressive (non-AR) methods without semantics distillation. It is natural to question whether incorporating semantics distillation could further improve its performance.

To investigate this, we introduce an auxiliary semantics-distillation loss into our tokenizer training pipeline. Specifically, we implement the latent representation projector as a three-layer MLP, similar to REPA (Yu et al., 2025b). The noised latent embeddings (`z_noised` at line 18 in Algorithm 1) are projected through the following layers: `Linear(16, 2048)` $\rightarrow$ `SiLU()` $\rightarrow$ `Linear(2048, 2048)` $\rightarrow$ `SiLU()` $\rightarrow$ `Linear(2048, 768)`, where 16 represents the latent embedding dimension and 2048 the intermediate projection dimension. This projector is trained to maximize the cosine similarity between the projected embeddings and semantic features extracted from the pretrained DINOv2-Base model (Oquab et al., 2024).

We have also experimented with a decoder-based semantics-distillation approach, *i.e.*, using an auxiliary Transformer-based decoder to predict pretrained features. After training both implementations for 50 epochs, we found their results were very similar, though the decoder-based implementation required more computation because of the decoder overhead. Thus, we adopted the simpler MLP-based approach for efficiency.

The loss coefficient for minimizing the cosine distance is set to $1.0$ for simplicity, which we do not sweep or tune. We train this tokenizer using the identical settings as the "+Extended" protocol described in Table 2: 200 epochs in total, activating GAN loss from epoch 100.

The results from this experiment (summarized in Table 4) demonstrate substantial improvements, particularly for diffusion-based non-AR models (e.g., SiT). With semantics distillation, our $l$-DeTok achieves state-of-the-art FID scores for non-AR models, clearly surpassing previous semantics-distilled tokenizers such as MAETok (Chen et al., 2025a) and VA-VAE (Yao & Wang, 2025). While AR models benefit relatively less from distillation—and performance may even slightly degrade compared to the non-distilled $l$-DeTok—their generative quality remains notably better than previous approaches.

Nonetheless, we emphasize the practical significance of our original $l$-DeTok formulation without semantics distillation, especially for domains like video, audio, proteins, poses, or trajectories, where strong pretrained semantic models comparable to DINOv2 (Oquab et al., 2024) may be unavailable or significantly weaker. Thus, our proposed tokenizer remains broadly valuable both with and without semantic distillation.

## A.5   CONVOLUTIONAL TOKENIZER

**Model.** We reuse MAR-VAE and SD-VAE architectures with a downsampling ratio of 16 and latent dimension 16, producing latent representations of shape $(16, 16, 16)$ for $(C, H, W)$. We add only a *few lines* of latent-denoising code (similar to Algorithm 1) to CNN-based tokenizers. Tokenizers are trained for 50 epochs, both with and without latent-denoising, using hyperparameters identical to our Transformer-based tokenizers, potentially disadvantaging CNN models.[3] We enable GAN loss from epoch 10 for CNN tokenizers. We do not sweep the noise standard deviation and only run with $\gamma = 3.0$ and $\gamma = 0.0$ (baseline). Further hyperparameters and noise strength tuning and longer training may yield additional improvements for CNN-based tokenizers, which we leave for future work.

**Results.** We report FID@50k under best CFG scales. With the default CNN baseline, MAR-B achieves 3.32 FID and SiT-B 7.11 FID. With our denoising-based CNN tokenizers, MAR-B improves to 2.82 FID and SiT-B to 5.62 FID, demonstrating that $l$-DeTok provides consistent gains also in convolutional networks. Our results clearly demonstrate the general, architecture-agnostic nature of our latent-denoising idea.

## A.6   1D TOKENIZERS

**Implementations.** We construct 1D continuous tokenizers by minimally modifying the 2D continuous tokenizers used in our main experiments. Concretely, we keep the image encoder and decoder backbones unchanged and only alter the latent parameterization and positional embeddings. Following Yu et al. (2025a), we append $K \in \{32, 64, 128\}$ learnable latent tokens to the encoder input sequence, leading to $256 + K$ tokens for a $256 \times 256$ image with patch size 16 (*i.e.*, 256 patch tokens plus $K$ latent tokens). After encoding, only the $K$ latent tokens are passed to the decoder. The decoder then uses learnable mask tokens to reconstruct the original images from these $K$ latents. All Transformer blocks are based on self-attention. For simplicity, we drop RoPE and use only learnable absolute positional embeddings.

**1D tokenizers with $l$-DeTok.** When applying $l$-DeTok to 1D tokenizers, we use interpolative latent noising with $\gamma = 2.0$ and omit masking entirely for simplicity. All other training hyperparameters (optimizer, schedule, batch size, and loss) are kept identical to those used for our 2D continuous tokenizers, which is convenient for comparison but may be suboptimal for the 1D tokenizers.

---

[3]In our experience, Transformer- and CNN-based tokenizers typically require distinct hyperparameter sets.

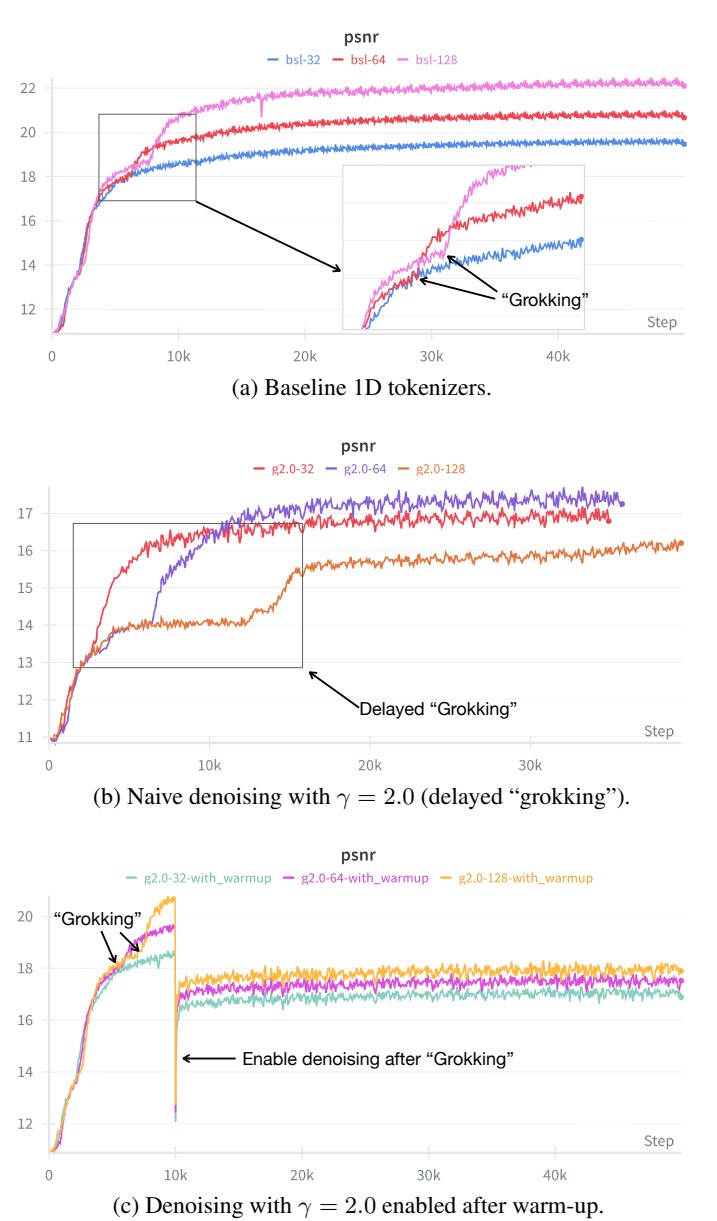

Figure A.3: **Training dynamics of 1D tokenizers.** PSNR curves for different token counts $K$. (a) Baseline 1D tokenizers exhibit a "grokking"-like phase where PSNR suddenly improves after a plateau. (b) Directly adding denoising with $\gamma = 2.0$ delays this transition, especially for larger $K$. (c) Enabling denoising only after the baseline grokking phase (10-epoch warm-up) yields stable training across all $K$. X-axis: calibrated global step, 1k-step denotes 1 ImageNet epoch.

**Grokking.** We observe a practical "grokking"-like phase when training 1D tokenizers: AA (Fig. A.3a). This behavior already appears in the *baseline* 1D tokenizers without $l$-DeTok. A naive application of denoising with $\gamma = 2.0$ (without any warm-up) further delays this transition, especially for larger $K$ (Fig. A.3b), so under a fixed training budget the tokenizers remain undertrained and yield poor downstream FID. To mitigate this, we warm up the 1D tokenizer for 10 epochs by disabling the denoising task, and then enable it from epoch 10 onward with a milder noising level $\gamma = 2.0$ to reduce task difficulty.[4] This staged schedule makes all runs stable and successful (Fig. A.3c), and is analogous to common tokenizer training practices that warm up the model before

---

[4]We initially used $\gamma = 3.0$ and observed poor downstream performance. We first attributed this failure to the large $\gamma$, but later found that the delayed grokking was the primary cause.

turning on GAN or perceptual losses. We hope this full transparency will serve as a useful reference and provide concrete data points for future research.

### A.7 Vector-quantized (VQ) Tokenizers

**Implementations.** For the VQ tokenizer, we start from the same continuous tokenizer architecture and only replace the bottleneck head. In the continuous variant, a linear layer maps Transformer tokens from the model width to the latent channels (*e.g.*, 16 or $2 \times 16$ for the variational autoencoder). In the VQ variant, this head is replaced by a VQ bottleneck: a linear projection first maps each token to a 64-dimensional latent vector, which is then quantized using a codebook of size 4096, *i.e.*, a learnable matrix of shape $(4096, 64)$. Following prior work (Yu et al., 2021; 2025a), we $\ell_2$-normalize the codebook vectors and train with the standard VQ objective (reconstruction loss plus codebook loss) and a commitment loss with weight 0.25; we refer readers to these works for further formulation details. All other components of the encoder and decoder are unchanged. As with our 1D and continuous tokenizers, we reuse the same training hyperparameters (optimizer, schedule, and loss weights apart from the VQ terms), which simplifies comparison but may be suboptimal for the VQ setting.

**Generators.** On top of the learned VQ tokenizer, we train discrete auto-regressive (AR) generators in two settings: (i) *RandomAR*, where the 2D grid of discrete tokens is visited in a random permutation at both training and sampling time; and (ii) *RasterAR*, where the same architecture is trained with the standard raster-scan ordering. We directly use the implementation from RAR (Yu et al., 2024) without modification and fix the generation order (random or raster-scan) consistently during both training and inference. These models are decoder-only, causal-attention Transformers, trained with cross entropy loss in a teacher-forcing manner.

**VQ tokenizers with $l$-DeTok.** Our $l$-DeTok variant for VQ tokenizers is intentionally minimal: we only leverage masking as deconstruction during training. Starting from the baseline RasterAR setup, we sweep masking ratios of 0.7 and 0.95. Both ratios yield better downstream FID than the baseline VQ tokenizer, with 0.95 performing best. We therefore adopt a masking ratio of 0.95 for RandomAR runs.

## B Additional Results

### B.1 Effect of Noising on Reconstruction Performance

Figure B.1 reports tokenizer reconstruction quality (measured by rPSNR and rFID) under various noising strategies discussed in Section 5.1. All tokenizers are trained for 50 epochs without GAN loss. As expected, reconstruction performance degrades with increased noise strength (noise standard deviation or masking ratio). Nevertheless, even under substantial noise levels, our tokenizers achieve reasonably strong reconstruction quality, consistently exceeding 24.0 rPSNR and remaining below 1.2 rFID.

Figure B.1-(b) highlights the importance of the additional 10-epoch decoder fine-tuning step for constant-masking tokenizers, which effectively mitigates the visual degradation and photorealism loss stemming from the training-inference distribution mismatch. This fine-tuning significantly improves decoder reconstruction, particularly at higher masking ratios. Training with randomized masking consistently outperforms constant masking, indicating that variable masking ratios lead to more robust reconstruction and superior generative outcomes.

### B.2 FID *vs.* CFG curves.

Figures B.2 and B.3 compare how classifier-free guidance (CFG) scales influence generative performance (FID and IS) across different tokenizers and generative models. We use warm colors to denote semantics-distilled tokenizers (marked by "*"), and cool colors for tokenizers without distillation. Our $l$-DeTok consistently achieves stronger performance across varying CFG scales, improving notably over standard tokenizers and matching or even surpassing semantics-distilled ones. Final FID scores computed over 50,000 generated images at optimal CFG scales are summarized in Table 4.

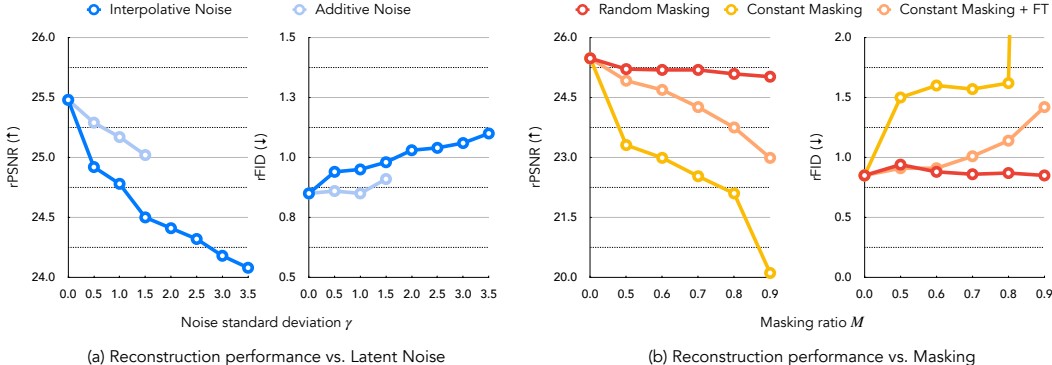

Figure B.1: **Tokenizer reconstruction under various noising strategies.** We report rPSNR and rFID for tokenizers studied in Section 5.1, trained for 50 epochs without GAN loss. With masking ratio $M = 0.9$, constant masking unexpectedly collapses performance, yielding an unexpected rFID of 13.95 (out of plot range).

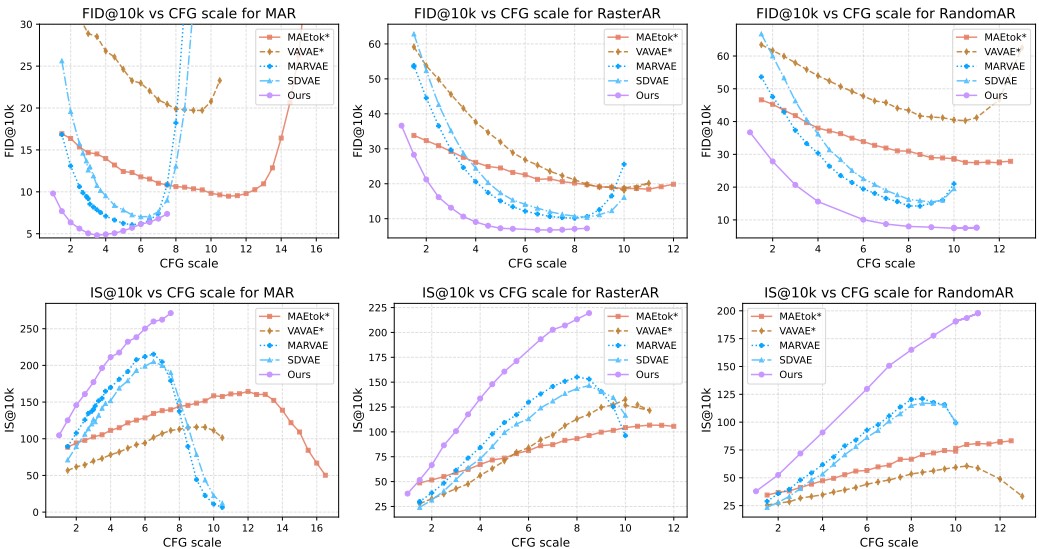

Figure B.2: **Tokenizer comparison for autoregressive models.** We report FID@10k and IS@10k scores at different classifier-free guidance (CFG) scales. Semantics-distilled tokenizers (denoted by "*") are shown in warm colors, while those without distillation are shown in cool colors. Our $l$-DeTok consistently achieves superior FID and IS metrics compared to other tokenizers. See Table 4 for results on 50,000 images evaluated at the optimal CFG scales.

## B.3 TRAINING CURVES

**Convergence.** We visualize the training loss of our tokenizers under different latent denoising strategies, *i.e.*, tokenizers we use in Sections 5.1.1 and 5.1.2. Figures B.5 and B.4 vary only the masking ratio and the noise standard deviation $\gamma$, respectively, while keeping all other hyperparameters fixed. In both cases the curves decrease smoothly and follow the same convergence pattern, with heavier corruption leading to higher final loss but no signs of instability under this setting.

**Baseline vs. $l$-DeTok.** Figure B.6 compares full training curves of the baseline tokenizer and $l$-DeTok under the same 200-epoch schedule, with GAN enabled from epoch 100, leading to sudden jumps of loss. The per-step compute is effectively unchanged, and both models converge at similar speed in both the autoencoding and adversarial stages. Although $l$-DeTok optimizes a harder denoising problem and therefore attains a higher loss, this does not slow optimization, while consistently

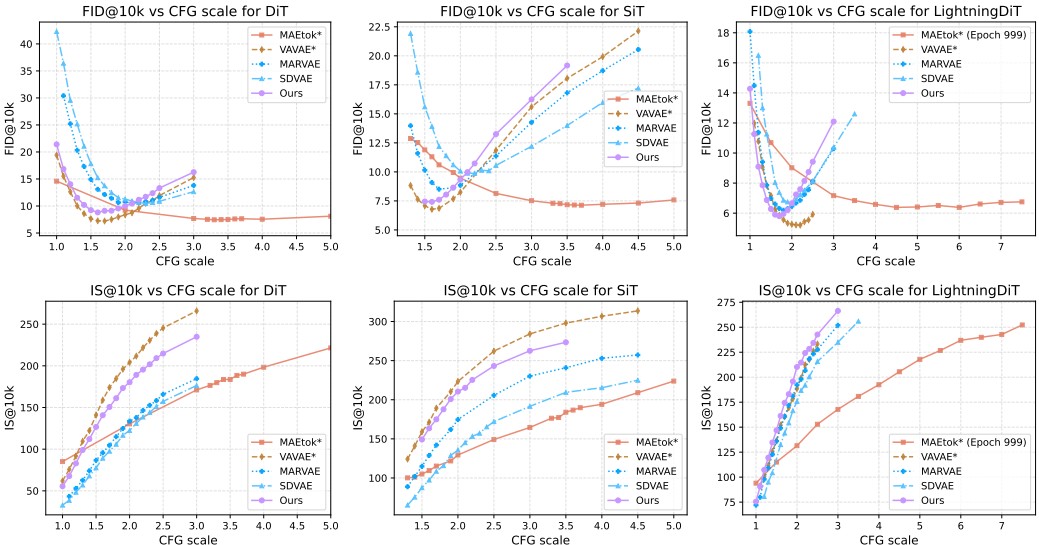

Figure B.3: **Tokenizer comparison for non-autoregressive models.** We report FID@10k and IS@10k scores at different classifier-free guidance (CFG) scales. Semantics-distilled tokenizers (denoted by "*") are shown in warm colors, while those without distillation are shown in cool colors. Our $l$-DeTok consistently outperforms standard (non-semantics-distilled) tokenizers, matching the performance of the best semantics-distilled tokenizer (VA-VAE (Yao & Wang, 2025)) and surpassing MAETok (Chen et al., 2025a) in IS. See Table 4 for results on 50,000 images evaluated at the optimal CFG scales.

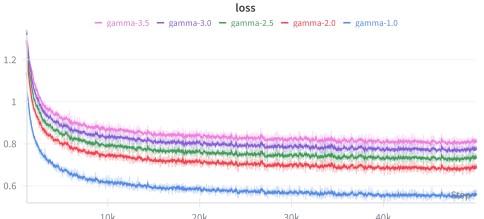

Figure B.4: Training loss under different noise amplitudes $\gamma$. Larger $\gamma$ increases the reconstruction difficulty and thus the final loss, but all settings converge monotonically without instability under this setting. We adopt $\gamma = 3.0$ in final experiments, which slightly increases loss but yields the best downstream generative quality. X-axis: calibrated global step, 1k-step denotes 1 ImageNet epoch.

Figure B.5: Training loss under different masking ratios, with all other hyperparameters fixed. Heavier masking (*e.g.*, 0.7–0.95) makes reconstruction harder and yields slightly higher final loss. All curves are smooth and stable, indicating that adding the denoising branch does not introduce optimization issues under this setting. X-axis: calibrated global step, 1k-step denotes 1 ImageNet epoch.

improving downstream FID/IS. Together with Table A.2, this confirms that the denoising objective introduces negligible training-time overhead relative to a vanilla tokenizer.

**Scalability.** Finally, Figure B.7 examines how the tokenizer training loss scales with model size. Larger tokenizers achieve lower absolute loss, as expected. More importantly, after normalizing each curve by its initial loss, the baseline and $l$-DeTok trajectories remain closely aligned at all scales, with only minor deviations for the smallest (SS) variant, indicating similar convergence behavior. These curves suggest that the proposed denoising principle extends smoothly to larger tokenizers without introducing additional optimization barriers or compute bottlenecks.

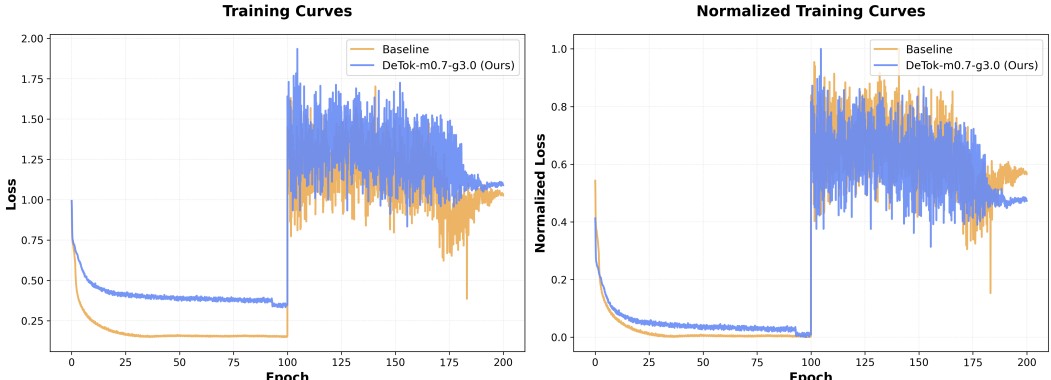

Figure B.6: Full training curves of the baseline tokenizer and $l$-DeTok under the same schedule (200 epochs, with GAN enabled starting at epoch 100). The per-step wall-clock time is essentially identical. Both models converge quickly in the autoencoding stage and remain stable throughout adversarial training. $l$-DeTok shows a higher loss due to the harder denoising task, but reaches its plateau in a similar number of epochs while enabling better downstream FID/IS.

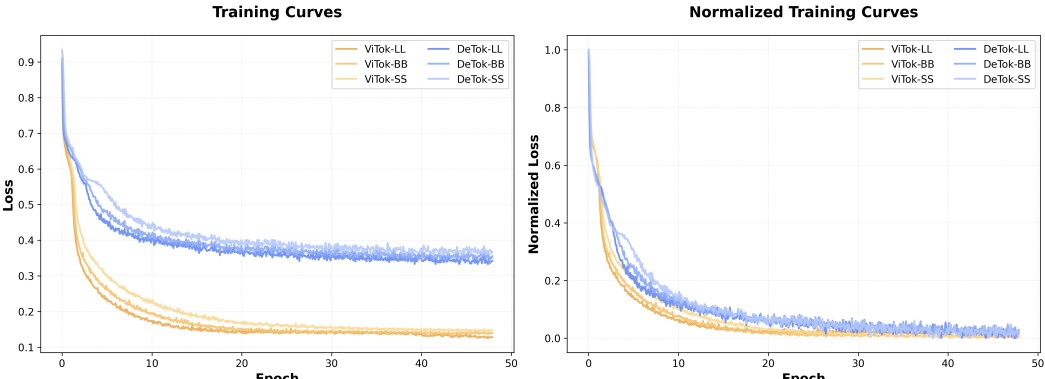

Figure B.7: Scaling behavior of baseline and $l$-DeTok tokenizers across model sizes (ViTok-SS/BB/LL). Left: absolute loss, where larger models achieve lower final loss as expected. Right: loss normalized into [0, 1]. At each scale, the normalized trajectories of baseline and $l$-DeTok nearly resemble each other, indicating similar convergence speed and optimization difficulty. This shows that our denoising objective scales smoothly without introducing optimization or compute bottlenecks under this setting. ViTok: baselines. DeTok: ours with $\gamma = 3.0$ and $M = 0.7$.

## C  TEXT-TO-IMAGE GENERATION

We further validate $l$-DeTok on text-to-image (T2I) generation. Following the protocol in (Bao et al., 2022; Yu et al., 2025b), we train models from scratch on the MS-COCO train split (Lin et al., 2014) and evaluate on the validation split. To adapt SiT and MAR for T2I, we make minimal modifications: (1) For SiT, we concatenate text and image tokens, computing the loss only on image tokens. The AdaLN modulation input is simplified from {timestep embedding + class embedding} to only timestep embedding. (2) For MAR, we fill the buffer tokens, which originally stores class embeddings, with text embeddings. In both cases, text embeddings are extracted using CLIP-L as in Bao et al. (2022); Yu et al. (2025b).

We train T2I variants of MAR-B and SiT-B under identical settings with different tokenizers. Except for SDVAE, all tokenizers are pretrained on ImageNet $256 \times 256$ without any tuning on the MSCOCO dataset. MAR models are trained for 800 epochs while SiT models are trained for 1000 epochs.

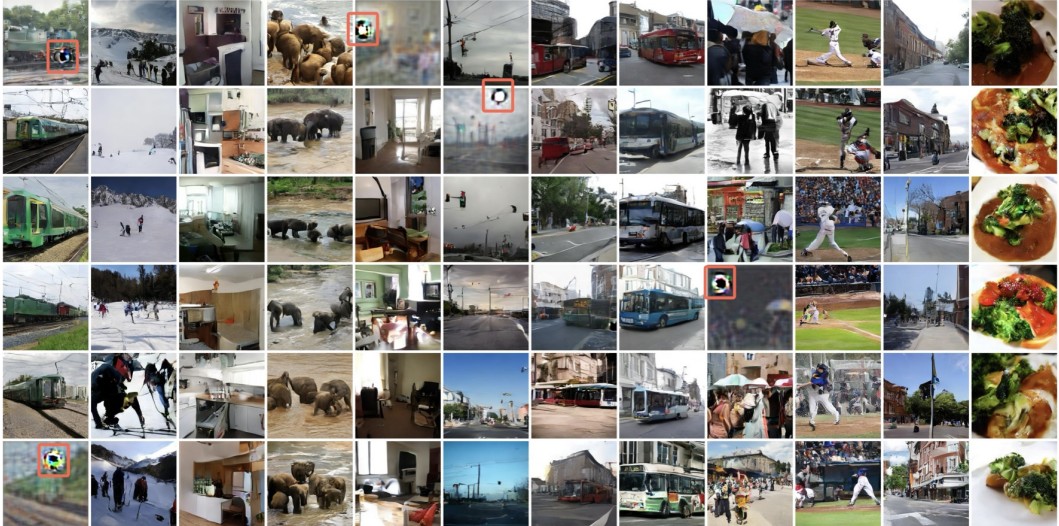

Figure C.1: **Text-to-image generation using MAR-VAE tokenizer.** Uncurated text-conditional generations on MS-COCO with T2I-MAR-B trained using MAR-VAE tokenizer (Li et al., 2024a). Red boxes highlight the "spot artifacts" reported in previous works (Fan et al., 2025; Team et al., 2025), which notably do not appear when using our tokenizer (see Figure C.4 for comparison).

**Qualitative Text-to-Image Comparison.** Figures C.1, C.2, C.3, and C.4 visualize uncurated text-to-image samples generated by T2I-MAR with various tokenizers. Other tokenizers frequently produce "spot artifacts" as previously reported (Fan et al., 2025; Team et al., 2025), whereas such artifacts are notably absent with our *l*-DeTok.

Text prompts for each column (from left to right) are: "A green train is coming down the tracks.", "A group of skiers are preparing to ski down a mountain.", "A small kitchen with a low ceiling.", "A group of elephants walking in muddy water.", "A living area with a television and a table.", "A road with traffic lights, street lights and cars.", "A bus driving in a city area with traffic signs.", "A bus pulls over to the curb close to an intersection.", "A group of people are walking and one is holding an umbrella.", "A baseball player taking a swing at an incoming ball.", "A city street lined with brick buildings and trees.", and "A close up of a plate of broccoli and sauce."

## D  LIMITATION

**Limitations.** On the theoretical side, our understanding of why *l*-DeTok works as well as it does is still limited: its success is supported by consistent empirical gains, but a principled and theoretical analysis of the underlying optimization dynamics and representation properties is left open. On the technical side, we observe a training/inference discrepancy in our tokenizer: the decoder is primarily trained on noise-injected latent embeddings, whereas it operates on almost noise-free embeddings during inference. Fine-tuning the decoder on clean latent embeddings partially addresses this discrepancy. Further investigation into mitigating this discrepancy could yield additional improvements. Nonetheless, the core motivation and insights of our work remain robust.

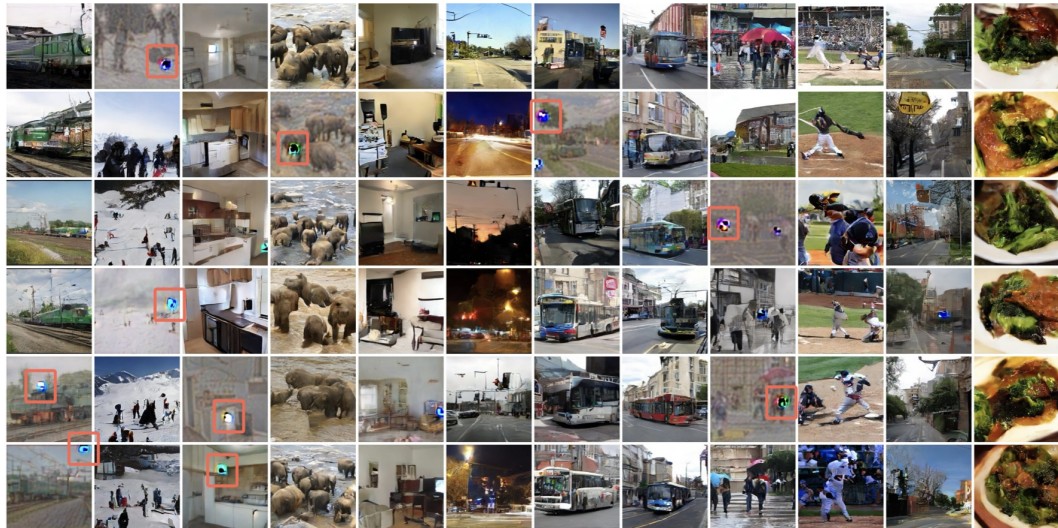

Figure C.2: **Text-to-image generation using SD-VAE tokenizer.** Uncurated text-conditional generations on MS-COCO with T2I-MAR-B trained using SD-VAE tokenizer (Rombach et al., 2022). Red boxes highlight the "spot artifacts" reported in previous works (Fan et al., 2025; Team et al., 2025), which notably do not appear when using our tokenizer (see Figure C.4 for comparison).

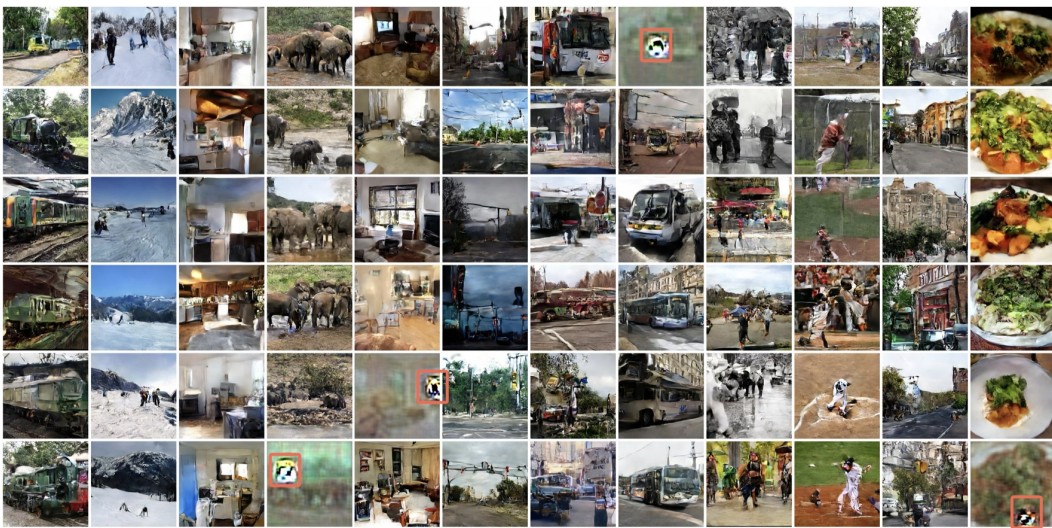

Figure C.3: **Text-to-image generation using VA-VAE tokenizer.** Uncurated text-conditional generations on MS-COCO with T2I-MAR-B trained using VA-VAE tokenizer (Yao & Wang, 2025). Red boxes highlight the "spot artifacts" reported in previous works (Fan et al., 2025; Team et al., 2025), which notably do not appear when using our tokenizer (see Figure C.4 for comparison).

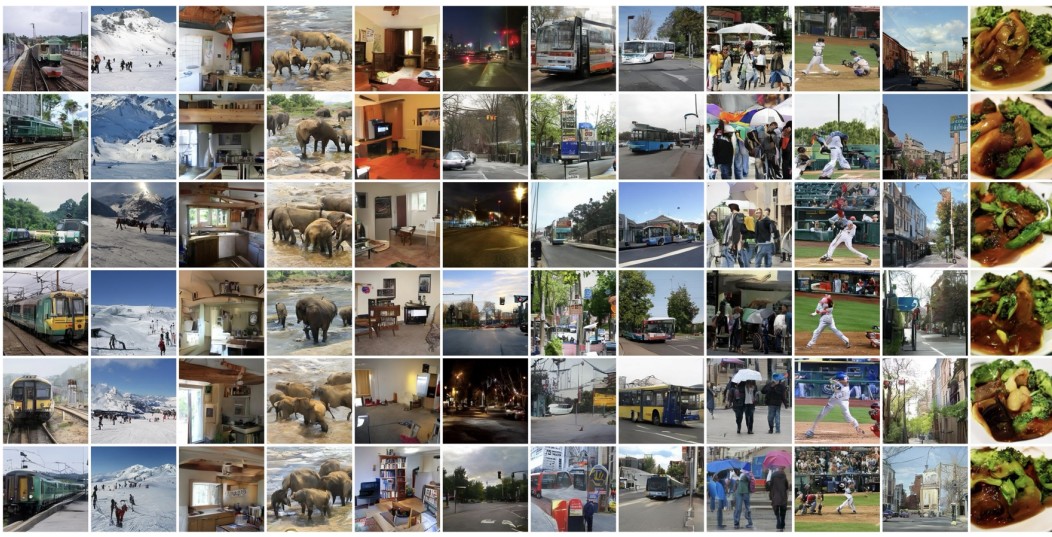

Figure C.4: **Text-to-image generation using our $l$-DeTok tokenizer.** Uncurated text-conditional generations on MS-COCO with T2I-MAR-B trained using our $l$-DeTok tokenizer (Yao & Wang, 2025).

