# OpenReview forum: "Latent Denoising Makes Good Tokenizers"
_ICLR.cc/2026/Conference — ICLR 2026 Poster_

### Official Review · Reviewer_pDBk · 2025-10-29

**Soundness:** 4
**Presentation:** 4
**Contribution:** 4
**Rating:** 8
**Confidence:** 4

**Summary:**

This paper focuses on training more effective visual tokenizers for subsequent generative modeling. Motivated by the shared denoising nature of current generative models, it incorporates the denoising objective into tokenizer training to align its embedding with downstream generative models. Specifically, this work introduces Latent Denoising Tokenizer (l-DeTok), a simple yet highly effective tokenizer trained to reconstruct clean images from latent embeddings corrupted via interpolative noise or random masking. Comprehensive experiments on various settings and types of generative models validate the effectiveness and generalization of the trained tokenizers.

**Strengths:**

1. This paper is well-motivated and easy to follow. It aligns tokenizer embeddings with the downstream denoising objective in generative modeling.
2. The writing of this paper is clear, and the expressions are polished and elegant. I appreciate this point.
3. The proposed method and spirit are general and robust, which is applicable to a wide spectrum of generative models.
4. The paper provides comprehensive experiments and ablations to validate the superiority of the proposed method.

**Weaknesses:**

1. How much additional computational overhead does the proposed denoising mechanism introduce compared with conventional tokenizer training? Detailed comparisons should be presented.
2. How do the performance gains vary across various tokenizer and generative model sizes? (results with different sizes of both tokenizer and generators)

**Questions:**

See weaknesses.

---

> ### Author Response · Authors · 2025-11-20
> **Response to Reviewer pDBk**
>
> We sincerely thank the reviewer for the thoughtful and extremely positive assessment of our work, and for highlighting its clarity, motivation, generality, and experimental thoroughness!
>
> Below we address the two main concerns and include compact tables to summarize the new results added in the revision.
>
> ### **W1. Computational overhead.**
>
> Our method introduces *no meaningful additional computational overhead* compared to conventional tokenizer training. Architecturally, we do not add any extra network components; we only change the behavior in the bottleneck layer by injecting latent noise (via interpolative noise and/or masking). This keeps the forward and backward passes unchanged up to negligible operations for sampling/interpolation. In the revised manuscript, we now report detailed training-time measurements in Table A.2 (page 16) and show training curves in Sec. B.3 (Figs. B.4–B.7). The wall-clock training time of our tokenizers closely matches that of the baseline (Table A.2), shown below:
>
> **Table A.2: Training cost.** Comparison of training schedules and approximate compute for different tokenizer and setting combinations. Models are trained on mixed hardware (A100, H100, A6000); we report approximate A100-equivalent hours over the full run. “Size” is Encoder–Decoder (S: Small, B: Base).
>
> | Size | Setting              | Epochs | Batch size | GAN start epoch | Approx. A100 hours |
> |------|----------------------|--------|------------|-----------------|--------------------|
> | S–B  | Baseline             | 50     | 1024       | disabled        | 180                |
> | S–B  | l-DeTok              | 50     | 1024       | disabled        | 160–180            |
> | B–B  | Baseline             | 200    | 1024       | 100             | 1150               |
> | B–B  | l-DeTok              | 200    | 1024       | 100             | 1140               |
> | B–B  | l-DeTok + Distill.   | 200    | 1024       | 100             | 1900               |
> | S–B  | 1D Baseline          | 50     | 1024       | disabled        | 170–190            |
> | S–B  | 1D l-DeTok           | 50     | 1024       | disabled        | 170–190            |
> | S–B  | VQ Baseline          | 50     | 1024       | disabled        | 190                |
> | S–B  | VQ l-DeTok           | 50     | 1024       | disabled        | 170                |
>
> ---
>
> ### **W2. Performance across tokenizer and model sizes.**
> This is a great point! We agree that understanding scalability with respect to different components is important, and we have expanded our experiments accordingly in the revision. Specifically, Table 3 now reports results for multiple generator sizes (SiT-B/L/XL and MAR-B/L), all trained on top of a fixed base-sized tokenizer, comparing the baseline tokenizer to our l-DeTok under identical training epochs. Across all scales we have experimented with, l-DeTok consistently improves FID. For example, on ImageNet 256×256:
>
> **Table 3: Scalability.**
> | Model   | FID ↓ with baseline | FID ↓ with l-DeTok |
> |---------|---------------------|--------------------|
> | SiT-B   | 7.08                | **5.13**         |
> | SiT-L   | 4.66                | **3.49**         |
> | SiT-XL  | 4.47                | **3.14**         |
> | MAR-B   | 3.65                | **2.43**       |
> | MAR-L   | 2.44                | **2.08**        |
>
> These trends persist when we extend training to 800 epochs in the system-level comparison (Table 5), indicating that the benefits of l-DeTok are stable across both model capacity and training duration in the regimes we have tested.
>
> Due to limited compute and time, we have not yet completed a full sweep over differently sized tokenizers (e.g., smaller/larger SS and LL variants) or more compute-intensive generative model variants such as MAR-H, but we provide initial training curves for different tokenizer sizes in Fig. B.7. We plan to run these larger-scale and longer-duration experiments as more compute becomes available, and we expect them to further clarify the scaling behavior of l-DeTok.
>
> We have also updated the discussion section (Sec. 6) to explicitly frame scalability to larger models and datasets as a central and promising direction for future work.
>
> ---
> We hope these additions and clarifications address the reviewer’s concerns.
>
> We are happy to elaborate further if needed and sincerely appreciate the time and insight the reviewer brought to this review!

---

> > ### Comment · Reviewer_pDBk · 2025-11-26
> >
> > Thanks for the solid reply, which has addressed most of my concerns about scalability and computational overhead. My final rating is accept.

---

### Official Review · Reviewer_gSib · 2025-11-01

**Soundness:** 3
**Presentation:** 3
**Contribution:** 2
**Rating:** 6
**Confidence:** 4

**Summary:**

The paper proposes Latent Denoising Tokenizer (l-DeTok), a visual tokenizer trained to reconstruct clean images from corrupted latent embeddings using interpolative Gaussian noise and random masking during tokenizer training. The authors demonstrate that it leads to consistent gains for both non-autoregressive and autoregressive models. Extensive experiments on ImageNet and MS-COCO across two resolutions support the claim that a simple, intuition-driven denoising objective can yield broadly useful tokenizer embeddings that transfer across generative paradigms.

**Strengths:**

1. Generalizes across AR and non-AR generators.
- The same tokenizer improves both diffusion-based (non-AR) and autoregressive models without architectural changes, indicating that the denoising-aligned latent space is broadly compatible with diverse generation mechanisms.

2. Simple and practical method such that no external encoder alignment or semantic distillation required.
- While recent approaches emphasize semantics distillation from powerful pretrained vision models, l-DeTok shows that a lightweight, corruption-robust training objective can match (or surpass) such methods without explicitly aligning to external encoders.

3. Drop-in usability and clear training signal.
- The method is easy to implement, and the paper provides a clear heuristic that practitioners can readily apply.

**Weaknesses:**

Regarding with Related work, please add the following references.
- Zhao et al., ε-VAE: Denoising as Visual Decoding.
- Tschannen et al., Generative Infinite-Vocabulary Transformers.
- Kim et al., Efficient Generative Modeling with Residual Vector Quantization-Based Tokens.

**Questions:**

1. Discrete extension of the method.
- Given that the corruption scheme includes random masking, which is naturally defined in discrete token spaces (masking or replacing code indices), how well would the approach extend to discrete tokenizers such as VQ or RVQ? A brief discussion of potential pitfalls, such as codebook collapse, corruption schedule design and any small-scale evidence may extend the scope of the contribution.


2. Orthogonality to aggressive compression and compact regimes.
- Recent work indicates that aggressive latent compression, such as approximately 32 tokens or adaptive-length tokenization, can improve both quality and efficiency. Is the proposed denoising-aligned training orthogonal to these compression strategies, and does combining them yield further gains? In particular, beyond the reported configuration with patch size 16 and latent dimension 16, are there results at more compact representation?

References
- Yu et al., 2024. An Image is Worth 32 Tokens for Reconstruction and Generation
- Duggal et al., 2025. Adaptive Length Image Tokenization via Recurrent Allocation

---

> ### Author Response · Authors · 2025-11-20
> **(1/2) Response to Reviewer gSib**
>
> We sincerely thank the reviewer for the detailed and thoughtful feedback. We are encouraged that you recognize the generality, simplicity, and practical utility of our proposed method. Below we address the requested additions to related work and the two main questions on discrete extensions and aggressive compression. All changes described here have been incorporated into the revised manuscript.
>
> **Related work.**
> We have added the suggested references (Zhao et al., *ε-VAE*; Tschannen et al., *Generative Infinite-Vocabulary Transformers*; Kim et al., *Efficient Generative Modeling with RVQ-Based Tokens*) and now discuss their relationship to our method in the revised related work section.
>
> ---
>
> ### **Q1. Discrete extension to VQ tokenizers**
>
> This is an excellent point, and it motivated us to implement and test a VQ variant of our method. In the revision, we add implementation details in Sec. A.7 and results in Table 9.
>
> In brief, we keep the tokenizer architecture unchanged and only replace the continuous bottleneck with a standard VQ head. Through the VQ head, tokens are projected to 64-dimensional vectors and quantized with a codebook of size 4096. Following prior work (Yu et al., 2021; Yu et al., 2025), we use ℓ₂-normalized codes and a commitment loss weight of 0.25. On top of this VQ tokenizer, we train two discrete AR generators from RAR (Yu et al., 2024): **RandomAR** (random ordering) and **RasterAR** (raster-scan ordering).
>
> Our l-DeTok VQ variant is intentionally minimal: we only use masking as the corruption mechanism during tokenizer training. We sweep masking ratios of 0.7 and 0.95 in the RasterAR setting and find 0.95 to work best (both variants are better than baseline VQ); we then adopt 0.95 for RandomAR without any tuning.
>
> As summarized below (Table 9 in the paper), l-DeTok improves discrete autoregressive models under both generation orders: for RandomAR, FID/IS improve from 8.48/144.93 to 7.03/162.68; for RasterAR, from 5.29/204.15 to 4.91/208.61.
>
> We did not observe codebook collapse in these runs.
>
> Overall, these results support that the *denoising* principle extends naturally to discrete tokenizers. We leave exploring other kinds of noise in the discrete space for future work.
>
> **Table 9: Vector-quantized tokenizers on ImageNet 256×256 (FID@50k / IS@50k, w/ CFG).**
>
> | Setting      | RandomAR-B FID↓ | RandomAR-B IS↑ | RasterAR-B FID↓ | RasterAR-B IS↑ |
> |-------------|-----------------|----------------|------------------|----------------|
> | Baseline VQ | 8.48            | 144.93         | 5.29             | 204.15         |
> | +Ours       | **7.03**        | **162.68**     | **4.91**         | **208.61**     |

---

> > ### Author Response · Authors · 2025-11-20
> > **(2/2) Response to Reviewer gSib**
> >
> > ### **Q2. Aggressive compression and compact 1D regimes**
> >
> > We agree that aggressive compression is an emerging and interesting regime to study. In response, we add 1D tokenizer experiments; Sec. A.6 presents implementation details and Table 8 shows the results.
> >
> > Concretely, we construct 1D continuous tokenizers by appending $K \in \{32, 64, 128\}$ learnable latent tokens to the encoder input (following Yu et al., 2025) and only passing these $K$ latents to the decoder. This directly leads to compact representations (down to 32 tokens) while keeping the backbone architecture fixed.
> >
> > When applying l-DeTok, we use interpolative latent noising with $\gamma = 2.0$ (no masking) and keep all other hyperparameters identical to those used for our 2D tokenizers.
> >
> > We observe that l-DeTok is largely orthogonal to compression and can be combined with it: for moderate and high token budgets, gains are clear and consistent.
> >
> > For example, at $K = 64$ and $K = 128$, both MAR and SiT see consistent FID improvements under both w/ and w/o CFG settings (Table 8).
> >
> > In the extremely compressed $K = 32$ regime, we see a mild trade-off: FID improves for the w/o CFG setting but only matches or slightly degrades under CFG setting.
> >
> > **Table 8: 1D continuous tokenizers on ImageNet 256×256 (FID@50k).**
> >
> > | #Tokens | Setting    | MAR-B FID↓ (w/ CFG) | MAR-B FID↓ (w/o CFG) | SiT-B FID↓ (w/ CFG) | SiT-B FID↓ (w/o CFG) |
> > |--------|------------|---------------------|----------------------|---------------------|----------------------|
> > | 32     | Baseline   | **5.22**            | 17.63                | **6.99**            | 18.91                |
> > | 32     | +Ours      | 5.51                | **14.87**            | 7.02                | **16.46**            |
> > | 64     | Baseline   | 4.58                | 17.73                | 7.05                | 22.82                |
> > | 64     | +Ours      | **4.12**            | **14.02**            | **6.40**            | **18.59**            |
> > | 128    | Baseline   | 3.92                | 18.63                | 7.95                | 28.36                |
> > | 128    | +Ours      | **3.14**            | **12.05**            | **6.20**            | **20.22**            |
> >
> > Moreover, in Sec. A.6, we report a “grokking”-like phenomenon when training these 1D tokenizers: the reconstruction loss can remain flat before undergoing a sharp improvement within just a few epochs.
> >
> > This behavior is quite different from 2D tokenizers. When combining l-DeTok, we show that a simple warm-up schedule (disabling denoising for the first 10 epochs) stabilizes training.
> >
> > Taken together, these results indicate that denoising-aligned training is compatible with recent compact-tokenization trends (e.g., Yu et al., 2024; Duggal et al., 2025): it can be layered on top of aggressive compression and continues to provide gains on these compact and non-structured representations.
> >
> > ---
> >
> > **Summary.**
> > In summary, we have
> > - (i) expanded the related work with the requested references and clearer positioning,
> > - (ii) added discrete VQ experiments that directly address the discrete-extension question, and
> > -  (iii) introduced 1D compact-token experiments that speak to compatibility with aggressive compression.
> >
> > We hope these additions clarify the scope and robustness of the proposed l-DeTok approach. We are happy to clarify or expand on any remaining questions.
> >
> > Once again, we sincerely appreciate the time and insight the reviewer brought to this review!
> >
> > ---
> > **References**
> >
> > 1. Yu et al., 2024. An Image is Worth 32 Tokens for Reconstruction and Generation
> >
> > 2. Duggal et al., 2025. Adaptive Length Image Tokenization via Recurrent Allocation
> >
> > 3. Yu et al., 2022. Vector-quantized Image Modeling with Improved VQGAN

---

> ### Comment · Reviewer_gSib · 2025-11-26
>
> Thank you for the thorough and constructive revision. I appreciate the substantial additional work that directly addresses the two main concerns raised in my initial review.
>
> 1. Discrete extension to VQ/RVQ-style tokenizers.
> - The added VQ experiments, along with the implementation details and consistent improvements under both RandomAR and RasterAR decoding, clearly demonstrate that the denoising-aligned training extends naturally to discrete latent spaces.
>
> 2. Compatibility with aggressive compression and compact regimes.
> - The new 1D tokenizer results convincingly show that l-DeTok remains orthogonal to aggressive compression and continues to offer benefits across a wide range of token budgets. The analysis of extreme compression settings and the training-stability observation (with warm-up) further clarify the method’s behavior in compact regimes.
>
> Beyond these additions, the strengthened related-work discussion and clarified methodological scope are appreciated. Overall, the authors have responded comprehensively and convincingly, and the new material meaningfully improves the submission. As these updates effectively address my comments, I am raising my score.

---

### Official Review · Reviewer_BHYG · 2025-11-01

**Soundness:** 3
**Presentation:** 4
**Contribution:** 3
**Rating:** 8
**Confidence:** 4

**Summary:**

This paper proposes a continuous VAE tokenizer that incorporates a denoising prior. The core idea is to corrupt the latent representations by injecting Gaussian noise or applying masking, and then training the tokenizer to reconstruct the original, uncorrupted latents. This design aims to address the discrepancy between the objectives of tokenizer training and subsequent image generative modeling, thereby improving the "denoisability" of the VAE latent space. The authors provide extensive experiments that effectively validate the proposed denoising tokenizer. Additionally, the paper presents several interesting empirical findings, including comparisons between random and fixed masking, as well as interpolative versus additive noise.

**Strengths:**

- The paper is well-motivated, addressing the critical challenge of aligning the training objectives of visual tokenizers and generative models.
- The proposed method is simple yet effective. The strategy of injecting interpolative or masking noise is conceptually sound and well-justified.
- The methodology and implementation details are presented with clarity, making the work easy to understand and reproduce.
- The experimental evaluation is extensive and well-structured, providing strong empirical support for the paper's claims.

**Weaknesses:**

- Convergence and scalability: A potential concern is the training convergence. While the denoising objective complements the pixel-reconstruction loss, it is plausible that learning to reconstruct from corrupted latents could slow down convergence compared to a vanilla baseline. It would be beneficial for the authors to provide an analysis of the training speed and computational overhead. Furthermore, a discussion on the scalability of the proposed method to larger models and datasets would strengthen the paper.

- Distribution of noise level: The paper appears to use a uniform distribution for the noise level factor \tau. Drawing inspiration from recent diffusion-based methods (e.g., SD3), which have shown that non-uniform timestep sampling can improve performance, it would be interesting to investigate whether a non-uniform sampling strategy for \tau could offer similar benefits for the proposed tokenizer.

**Questions:**

see weaknesses

---

> ### Author Response · Authors · 2025-11-20
> **(1/3) Response to Reviewer BHYG**
>
> We sincerely thank the reviewer for the thoughtful, constructive, and encouraging review. We are particularly grateful for your recognition of the paper’s motivation, clarity of presentation, and the simplicity and effectiveness of our proposed method. Below we address the two main concerns (W1–W2), and we have incorporated the corresponding analyses and results into the revised manuscript.
>
> ### **W1. Convergence, overhead, and scalability**
>
> **Convergence.**
>
> For our main 2D continuous and VQ tokenizers, we do **not** observe convergence issues when incorporating our method. In the revision, we now include training curves comparing the baseline tokenizer and our l-DeTok tokenizer across different model sizes and training schedules in Sec. B.3 (Figs. B.4–B.7). We plot both the raw loss and a normalized version (rescaled to [0,1]) to make trends comparable. In all cases, the convergence behavior of l-DeTok closely follows that of the baseline, without training instability.
>
> For full transparency, we also report a behavior we discovered during the rebuttal period for **1D tokenizers**, which can lead to training instability. Details are given in Sec. A.6. Specifically, even for the *baseline* 1D tokenizers (without l-DeTok), we observe a “grokking”-like phase: the reconstruction loss can stay nearly flat, or decrease very slowly, for many epochs and then suddenly drop over a small number of epochs (Figure A.3(a)). If we naively enable the denoising bottleneck from the beginning, this transition is delayed or suppressed, and performance degrades (Figure A.3(b)). To address this, we adopt a simple warm-up schedule: we train 1D tokenizers without enabling latent denoising for the first 10 epochs, and then enable it from epoch 10 onward. This is analogous to common practice in GAN-based tokenizer training, where one first pretrains with a simple reconstruction loss and only later turns on more challenging GAN loss. With this warm-up, the 1D models recover the grokking transition and converge reliably (Figure A.3(c)). We hope these observations, reported in the revision for completeness and transparency, will be useful data points for future research.
>
> **Training speed / computational overhead.**
>
> Our method introduces *no meaningful additional computational overhead* compared to conventional tokenizer training. Architecturally, we do not add any extra network components; we only change the behavior in the bottleneck layer by injecting latent noise (via interpolative noise and/or masking in the input space). This keeps the forward and backward passes unchanged up to negligible operations for sampling/interpolation. In the revised manuscript, we now report detailed training-time measurements in Table A.2 (page 16) and show training curves in Sec. B.3 (Figs. B.4–B.7). The wall-clock training time of our tokenizers closely matches that of the baseline (Table A.2), shown below:
>
> **Table A.2: Training cost.** Approximate A100-equivalent hours for different tokenizers and settings.
>
> | Size | Setting            | Epochs | Batch size | GAN start epoch | Approx. A100 hours |
> |------|--------------------|--------|------------|-----------------|--------------------|
> | S–B  | Baseline           | 50     | 1024       | disabled        | 180                |
> | S–B  | l-DeTok            | 50     | 1024       | disabled        | 160–180            |
> | B–B  | Baseline           | 200    | 1024       | 100             | 1150               |
> | B–B  | l-DeTok            | 200    | 1024       | 100             | 1140               |
> | B–B  | l-DeTok + Distill. | 200    | 1024       | 100             | 1900               |
> | S–B  | 1D Baseline        | 50     | 1024       | disabled        | 170–190            |
> | S–B  | 1D l-DeTok         | 50     | 1024       | disabled        | 170–190            |
> | S–B  | VQ Baseline        | 50     | 1024       | disabled        | 190                |
> | S–B  | VQ l-DeTok         | 50     | 1024       | disabled        | 170                |
>
> As seen above, l-DeTok matches the baseline in wall-clock time within measurement noise. The only clearly more expensive variant is *l-DeTok + semantics distillation*, which is expected as it adds an external teacher model pass.

---

> > ### Author Response · Authors · 2025-11-20
> > **(2/3) Response to Reviewer BHYG (Continuation of W1)**
> >
> > *(Continuation of W1)*
> >
> > **Scalability.**
> > We appreciate the reviewer for raising this point! We agree that a clearer scaling picture would strengthen the paper. In the revision, we add **new experiments on multiple generator scales** (Table 3), using a fixed base-sized tokenizer and varying the generator. All models are trained with identical epochs on either the baseline tokenizer or our l-DeTok. Across all scales we tested, l-DeTok consistently improves FID on ImageNet 256×256:
> >
> > **Table 3: Scalability.**
> >
> > | Model  | FID ↓ (baseline) | FID ↓ (l-DeTok) | ΔFID (baseline − l-DeTok) |
> > |--------|------------------|-----------------|---------------------------|
> > | SiT-B  | 7.08             | **5.13**        | -1.95                     |
> > | SiT-L  | 4.66             | **3.49**        | -1.17                     |
> > | SiT-XL | 4.47             | **3.14**        | -1.33                     |
> > | MAR-B  | 3.65             | **2.43**        | -1.22                     |
> > | MAR-L  | 2.44             | **2.08**        | -0.36                     |
> >
> > These trends persist when we extend training to 800 epochs in the system-level comparison (Table 5), indicating that the benefit of l-DeTok is stable across both model capacity and training duration in the current regime.
> >
> > We have also updated the discussion section (Sec. 6) to pinpoint **scaling to larger models and datasets** as a central and promising direction for future work. We quote Section 6 to here for completeness:
> >
> > > Simple principles that scale well are at the core of generative modeling. In this work, we have shown that a *surprisingly simple* principle, i.e., *denoising*-based tokenizer, already provides consistent benefits across a broad design space, including non-autoregressive and autoregressive generative models, 2D/1D/VQ tokenizers, and both raster and random generation orders, while adding almost no system complexity. Several questions remain open.
> > >
> > >*First*, our experiments are conducted within a moderate-scale regime ($\mathcal{O}(10^6)$ samples), where l-DeTok improves performance when we scale up within this envelope. A natural next step is to test it in the frontier regime of web-scale models and datasets ($\mathcal{O}(10^7)$–$\mathcal{O}(10^{10})$ samples), and more challenging settings such as video generation, and to understand how denoising-aligned tokenizers behave in those cases.
> > >
> > > *Second*, our results also parallel the recent trend of semantics distillation from pretrained encoders such as DINOv2. Distillation is effective when strong teachers exist and align well with the target domain, but it *may* become a bottleneck as we move to larger data and broader distributions that *surpass* the teacher’s coverage, or to domains without reliable teachers. In contrast, l-DeTok is self-contained and does not depend on external teachers. At scale, we hope such task-aligned tokenizers could provide a more flexible path forward.
> > >
> > > *Finally*, reconstructing clean inputs from noisy latents makes our tokenizer objective reminiscent of $x_0$-prediction in modern generative models. Understanding the boundary—and eventual unification—among *reconstruction*, *denoising*, and *generation* is an intriguing direction (e.g., Li et al., 2025). We hope this perspective will help guide future work on scalable generative models and the tokenizers that support them.

---

> > > ### Author Response · Authors · 2025-11-20
> > > **(3/3) Response to Reviewer BHYG**
> > >
> > > ### W2. Distribution of noise level $\tau$
> > >
> > > We appreciate the reviewer’s suggestion to investigate non-uniform sampling for the noise level $tau$, inspired by recent diffusion-based methods (e.g., SD3). Following the reviewer’s suggestion, in the revision, we add an ablation on the **noise schedule** (Table 1), comparing:
> > >
> > > - baseline $\tau = 0$ (no oise),
> > > - uniform sampling $\tau \sim \mathcal{U}(0, 1)$ (our default setting), and
> > > - several logit-normal schedules for $\tau$ with different mean values.
> > >
> > > **Table 1: Impact of noise level sampling.**
> > >
> > > | Sampling of $\tau$                          | FID $\downarrow$ (MAR-B) | FID $\downarrow$ (SiT-B) |
> > > |--------------------------------------------|--------------------------|--------------------------|
> > > | $\tau = 0$ (baseline)                      | 3.31                     | 6.97                     |
> > > | $\tau \sim \mathcal{U}(0, 1)$              | 2.77                     | 5.56                     |
> > > | $\mathrm{logit}(\tau) \sim \mathcal{N}(-0.8, 1)$ | 2.79              | 6.04                     |
> > > | $\mathrm{logit}(\tau) \sim \mathcal{N}(0, 1)$    | 2.77              | 5.93                     |
> > > | $\mathrm{logit}(\tau) \sim \mathcal{N}(0.8, 1)$  | **2.58**          | **5.44**                 |
> > >
> > >
> > > In summary, the new ablation confirms that:
> > > - **Uniform $\tau$** already gives strong improvements over the baseline.
> > > - **Non-uniform $\tau$** can yield additional gains when biasing toward higher noise levels, consistent with the reviewer’s intuition.
> > >
> > > ---
> > >
> > > Once again, we thank the reviewer for the constructive comments!
> > >
> > > We believe the added analyses on convergence, scalability, and noise scheduling further strengthen our paper and clarify the behavior and generality of our method.
> > >
> > > We hope these additions address the reviewer’s concerns, and we are happy to clarify or expand on any remaining questions.

---

### Official Review · Reviewer_71cM · 2025-11-03

**Soundness:** 3
**Presentation:** 3
**Contribution:** 3
**Rating:** 4
**Confidence:** 5

**Summary:**

This paper introduces the Latent Denoising Tokenizer, a visual tokenizer framework designed to explicitly align with the denoising objectives common to modern generative models. The authors propose training tokenizers to reconstruct images from latent representations subjected to strong interpolative noise and/or random masking, departing from traditional pixel-reconstruction VAE objectives. By doing so, $l$-DeTok aims to produce latent embeddings that are robust and reconstructable under significant corruption, which theoretically matches the requirements of downstream denoising-centric generative models. Empirical evaluation covers six prominent generative models—both autoregressive and non-autoregressive—across ImageNet and MS-COCO text-to-image settings, demonstrating that $l$-DeTok yields consistent improvements over standard, semantics-distilled, and convolutional tokenizers.

**Strengths:**

- The paper is motivated by an accurate and under-discussed observation: modern generative models, regardless of architecture, are fundamentally denoising systems. Training tokenizers with explicit latent corruption (interpolative noise, masking) is a clean conceptual shift that breaks with the tradition of mere pixel-wise autoencoding. This alignment is theoretically meaningful and empirically justified.

- The authors benchmark $l$-DeTok across a broad spectrum of generative models, on both class-conditional (ImageNet) and text-conditional (COCO) tasks. Baselines include state-of-the-art semantics-distilled tokenizers (e.g., VA-VAE, MAETok), standard VAE-style tokenizers, and convolutional tokenizers. These comparisons are fair, with well-matched training recipes and strong ablations.

- The authors provide clear, actionable analysis of the key components of $l$-DeTok. For instance, Figure 2 and Figure 3 dissect the importance of interpolative versus additive noise and the effects of constant versus randomized masking ratio on FID/IS. This isolates the impact of each design choice, giving both insight and reproducibility.

- Strong Generalization and Architectural-Agnosticism: The improvements hold for both Transformer-based and CNN-based tokenizers, as shown in Section A.5, and do not depend on external semantics distillation resources—important for domains lacking large vision encoders.

**Weaknesses:**

- Lack of Theoretical Analysis Regarding Optimality or Limitations: The empirical link between denoising-aligned tokenizers and improved downstream performance is clear, but the theoretical rationale is underdeveloped. For example, there is no formal analysis or proof of why interpolative over additive noise leads to strictly more robust or generative-friendly latents (as claimed in Section 5.1.1). While Figure 2 empirically demonstrates this, a mathematical discussion (e.g., in terms of mutual information or denoising risk) would strengthen the scientific value.

- Training/Test Distribution Discrepancy and Impact: Section A.3 (in APPENDIX) raises the training/inference mismatch issue: since the decoder is trained primarily on heavily corrupted inputs but, in deployment, must reconstruct from nearly clean latents, there is a risk of distribution shift. While decoder fine-tuning helps (see Figure A.1, Figure A.2), the paper might understate the downside: performance gains appear to hinge in part on decoder adaptability, not just the quality of the encoder's latents. The extent to which this is a fundamental limitation (versus an optimization artifact) is not fully explored.

- Potential Over-Claims on "Semantics Distillation Independence": The abstract and body at times appear to overemphasize $l$-DeTok’s superiority or independence compared to semantics distillation. However, Section A.4 shows that adding semantics distillation to $l$-DeTok further improves performance—especially for non-AR models—sometimes even surpassing pure denoising. This suggests the two are complementary rather than exclusive. The paper should be more circumspect in presenting $l$-DeTok as a replacement, rather than a supplement, to semantics-based approaches.

- The evaluation should be benchmarked against more powerful and contemporary VAEs, such as the SD3-VAE or Flux-VAE. The currently used SD-VAE is an outdated and underperforming baseline due to its 4-channel latent space. A more meaningful and fair comparison would be against a modern 16-channel VAE. This is a crucial point, as we have observed a concerning trend in recent tokenizer research where comparisons are made against the old SD-VAE to inflate perceived performance gains. Such a comparison is not a fair assessment of the proposed method's true capabilities.

**Questions:**

Please refer to Weaknesses

---

> ### Author Response · Authors · 2025-11-20
> **(1/2) Response to Reviewer 71cM**
>
> We thank the reviewer for the careful reading and constructive questions. We are glad that the denoising perspective and the broad empirical study are found useful. Below we address each concern in turn and describe changes we make in the revision.
>
> ### **W1. On theoretical understanding of interpolative noise**
>
> Our work is indeed primarily empirical, and we acknowledge that we do not have a formal optimality analysis for interpolative noise.
>
> Importantly, we do not intend to claim that interpolative noise is *strictly* superior to additive noise in a theoretical sense. In Section 5.1.1, our claims are based on the empirical trends in Figure 2. We already note in L298 that:
>
> > “Nonetheless, we observe the additive latent noise still improves MAR but not SiT.”
>
> This is meant to indicate that additive noise can also be beneficial, and we do not claim that interpolative noise is *strictly* better than additive noise.
>
> We agree that a more explicit discussion of this limitation would improve the paper. In the revision, we have:
>
> - Clarified in Section 5.1.1 that our conclusions about interpolative vs. additive noise are **empirical observations** within our setup (L297).
> - Acknowledged in the limitation section that the success of our work remains empirical and the theoretical justification of our method is an open problem for future work (Appendix D).
>
> We hope this makes our position clearer: our contribution here is to provide systematic empirical evidence that interpolative noise works well in practice for denoising-aligned tokenizers, and we view a deeper theoretical explanation as an interesting direction beyond the scope of this paper.
>
> ---
> ### **W2. Training/inference mismatch and the role of the decoder**
>
> We appreciate the reviewer’s careful attention to the role of the decoder. One misunderstanding we want to clarify is that we do **not** “understate the downside”. On the contrary, already at submission time we highlighted this distribution shift issue in Appendix A.3. We would like to clarify these here.
>
> **(1) Most reported gains do *not* use decoder fine-tuning.**
> All generalization experiments (Table 4 in the revised manuscript) are based on l-DeTok  **without** any decoder fine-tuning. Likewise, in the main ImageNet system-level comparison (Table 5), the strong FID scores are already achieved with the original decoder **before** fine-tuning:
> - MAR-B: FID improves from 2.31 (MAR-VAE) to **1.61** with l-DeTok, *without* decoder fine-tuning.
> - MAR-L: FID improves from 1.78 (MAR-VAE) to **1.43** with l-DeTok, *without* decoder fine-tuning.
>
> **(2) Encoder-side latent denoising is responsible for the majority of the gains.**
>
> In fact, adding **decoder fine-tuning on top of l-DeTok** yields only a small additional improvement (e.g., 1.61 → 1.55, 1.43 → 1.35). In other words, decoder tuning contributes only **~0.06-0.08** FID improvement. This is opposite to the impression that the gains “hinge” on decoder adaptability.
>
> We hope these clarifications address your concerns and we’re glad to provide more discussion if needed.
>
> ---
> ### **W3. Positioning relative to semantics distillation**
>
> We thank the reviewer for raising this point. We believe there is a slight misunderstanding of our intended message. Our goal is **not** to present l-DeTok as a replacement for semantics distillation, but to show that:
>
> 1. l-DeTok can be trained **without any external semantic teacher** and already performs strongly.
>
> 2. When a strong teacher (e.g., DINOv2) exists, semantics distillation remains highly beneficial and l-DeTok **is compatible** with it.
>
> We do not claim “semantics distillation independence” in the paper. In fact, we explicitly investigate their *compatibility* in Section A.4. As stated in the main text:
>
> > To further investigate *whether* our l-DeTok can benefit from semantics distillation, we incorporate an auxiliary semantics-distillation loss (details in Sec. A.4). Remarkably, this *privileged* version of our tokenizer achieves the best FID scores for non-AR models, surpassing *all* previous semantics-distilled tokenizers.
>
> This experiment is designed to test whether semantics distillation and denoising are **complementary**. The results show that they are: **non-AR models achieve their best FID when semantics distillation is added on top of l-DeTok**, outperforming previous semantics-distilled tokenizers.
>
> In the revision, we have softened the comparative language and avoided any phrasing that suggests that denoising “supersedes” semantics distillation.
>
> We hope this makes our position clear: l-DeTok is valuable both **without** semantics distillation (self-contained without external dependency) and **with** semantics distillation (when strong teachers are available). In the time where semantics distillation and representation alignment dominate the tokenizer design space, we hope our work can offer a distinct and complementary perspective.

---

> > ### Author Response · Authors · 2025-11-20
> > **(2/2) Response to Reviewer 71cM**
> >
> > ### **W4. Comparison to stronger VAEs**
> > We share the reviewer’s concern about the community trend of comparing only against the original SD-VAE. We follow reviewer’s suggestion and have included **SD3-VAE** (SD3.5 reuses SD3-VAE) as a modern 16-channel convolutional tokenizer baseline in Table 4.
> >
> > However, when plugged into the same ImageNet class-conditional generative models under a unified training recipe, SD3-VAE is **less competitive than SD-VAE** in terms of generative performance. We report results here for completeness:
> >
> > **Table 4: Tokenizers trained *without* semantics distillation**
> > | Tokenizer      | MAR (FID↓) | RandomAR (FID↓) | RasterAR (FID↓) | SiT (FID↓) | DiT (FID↓) | Light.DiT (FID↓) |
> > |----------------|-----------:|----------------:|----------------:|-----------:|-----------:|-----------------:|
> > | SD-VAE         | 4.64       | 13.11           | 8.26            | 7.66       | 8.33       | 4.24             |
> > | SD3-VAE        | 6.46       | 40.65           | 21.03           | 9.57       | 13.63      | 5.52             |
> > | MAR-VAE        | 3.71       | 11.78           | 7.99            | 6.26       | 8.20       | 3.98             |
> > | Our l-DeTok    | **2.43**   | **5.22**        | **4.46**        | **5.13**       | **6.58**       | **3.63**          |
> >
> > At first glance this may look surprising, but it is consistent with our main message: being a “stronger VAE” in terms of pixel-wise reconstruction does **not** guarantee better generative performance. SD3-VAE was designed and tuned mainly for reconstruction quality, at a time when it was not yet clear what properties make tokenizers effective for generative modeling. Indeed, it has the best rFID among all tokenizers. This aligns with the emerging observation in recent work that **better reconstruction does not necessarily yield better generation metrics** [1]. Our work points out *denoising* as a way for improving downstream generative models than reconstruction alone.
> >
> >
> > Regarding Flux-VAE, we agree it is another important contemporary tokenizer. Due to current compute limits we have not yet run the full Flux-VAE suite for Table 4. If additional compute becomes available, we will extend Table 4 to include Flux-VAE.
> >
> >
> > We hope this clarifies that we both (i) share the reviewer’s concern about SD-VAE-only baselines, and (ii) already address it by benchmarking against a modern 16-channel VAE (SD3-VAE), with results that support our core claim about denoising-aligned tokenizers.
> >
> > ---
> > Once again, we thank the reviewer for the detailed feedback and hope our responses address your concerns. We are happy to answer any further questions.
> >
> >
> > ---
> > ### Reference
> >
> > [1] Yao, Jingfeng, Bin Yang, and Xinggang Wang. "Reconstruction vs. generation: Taming optimization dilemma in latent diffusion models." In CVPR. 2025.

---

> ### Author Response · Authors · 2025-11-26
>
> Reviewer 71cM,
>
> Thank you again for your detailed reviews. We have addressed your points in our response and updated manuscript.
>
> With the discussion deadline approaching, we would like to briefly highlight the main changes we made in direct response to your concerns:
>
> - We clarified that the comparison between interpolative and additive noise is empirical in our setting, and we now state the lack of a full theoretical explanation as a limitation and leave it for future direction.
> - We clarified the training–inference mismatch and the role of the decoder, and showed that most of the FID gains come from encoder-side latent denoising rather than decoder fine-tuning.
> - We softened the comparative language and emphasized that denoising and semantics distillation are complementary; l-DeTok both works without a teacher and benefits further when a strong teacher is available. It’s complementary to semantics distillation rather than completely replacing it.
> - We added comparisons to a modern 16-channel tokenizer (SD3-VAE) under a unified recipe, and showed that our method is better than it for generative performance in our setting.
>
> We hope these changes address your concerns and make the contribution clearer. If you find the revisions satisfactory, we would appreciate it if you could consider updating your review. If anything remains unclear, we are happy to elaborate further.
>
> Best regards,
> Authors

---

### Author Response · Authors · 2025-12-01
**Summary of Revisions: Denoising as a new principle for tokenization**

Dear Reviewers, Area Chairs, and Program Chairs,

We sincerely thank you for the time and thoughtful feedback dedicated to our manuscript. We are especially grateful for the highly constructive comments and engagement, which led to **a strong consensus of 8, 8, and 8 (reviewers gSib, BHYG and pDBk)** achieved during the discussion phase, *prior to score resetting*.

We also appreciate that **Reviewer gSib** replies that we *“have responded comprehensively and convincingly, and the new material meaningfully improves the submission,”* and that **Reviewer pDBk** concludes our *“solid”* reply *“addressed most of the reviewer’s concerns about scalability and computational overhead.”*

The rebuttal period has been highly productive, motivating us to further verify and broaden the core insight of our work.


We briefly highlight three key outcomes that demonstrate the universality and impact of our Latent Denoising approach:

1. **A unified principle across tokenizer families (Reviewer gSib).** We show that latent denoising is not tied to a particular architecture or representation, but is a general principle that extends to both discrete and compact structures.
    - **Compact and non-structured 1D latents (Table 8).** Our method remains effective even with non-structured, aggressively compressed 1D tokenizers. **Reviewer gSib** notes that the new 1D results *“show that l-DeTok remains orthogonal to aggressive compression and continues to offer benefits across a wide range of token budgets.”*
    - **Discrete latents (Table 9).** We extend l-DeTok to vector-quantized tokenizers and improve discrete autoregressive models. As **Reviewer gSib** highlights, the VQ experiments *“clearly demonstrate that the denoising-aligned training extends naturally to discrete latent spaces.”*
    - **Universality.** Across *continuous* and *discrete* latents, *2D* and *1D* layouts, and both *Transformer* and *CNN* encoders, aligning the tokenizer objective with the generator’s denoising task yields consistent gains.

2. **Strong latents without external semantic teachers (Reviewer 71cM).** In a landscape increasingly dominated by semantics distillation (e.g., DINOv2), our approach offers a self-contained alternative:
    - **Superiority without teachers  (Table 4).** We outperform all non–distillation-based tokenizers, including modern SD3-VAE suggested by **Reviewer 71cM**.
    - **Self-contained.** We achieve this using only an internal denoising objective, without any external encoders.
    - **Compatibility  (Table 4).** Already at submission time, we showed that l-DeTok is compatible with semantics distillation. With semantics distillation, l-Detok outperforms all distillation-based tokenizers (Table 4). We further clarify that our goal is not to replace semantics distillation.
    - **Encoder-driven gains.** We further clarify that the main improvements come from *stronger latents produced by the encoder*, rather than from decoder adaptability, addressing the misunderstanding raised by **Reviewer 71cM**.
    - **Novelty.** In an era where semantics distillation prevails, our work offer a **new** and complementary **“think differently”** perspective. We hope this provides a scalable path for domains where powerful semantic teachers are unavailable (e.g., video, poses, audio, 3D), while remaining fully compatible with semantics distillation when *it is available*.

3. **Robust scalability and practical deployability (Reviewers BHYG and pDBk).** We show the scalability and the robustness of our approach.
    - **Scales with generative model size (Tables 4 and 5).** We show that the gains from our tokenizer persist from base to large and XL generators across non-autoregressive (SiT) and autoregressive models (MAR), and under longer training (**Reviewers BHYG and pDBk**).
    - **Negligible overhead (Table A2).** We show that the training cost and convergence behavior of our tokenizers closely match conventional tokenizers. Our method is simple and adds no extra network components. As **Reviewer pDBk** summarizes, our reply *“addressed most of the reviewer’s concerns about scalability and computational overhead.”*
    - **Stable across noise schedules (Table 3).** We demonstrate that simple uniform noise already works well, and high-noise-biased schedules yield further improvements (**Reviewer BHYG**).

**Remark.** Current generative models are denoisers; their tokenizers should be too. Our work takes a different view in an era dominated by semantics distillation: we close this objective mismatch and show that **latent denoising is a simple, universal design principle that improves a wide range of tokenizers and generators with essentially negligible system cost.**

We sincerely appreciate the strong and constructive support from Reviewers BHYG, pDBk, and gSib, and we hope the revised manuscript would provide more insights for future research within and beyond the design of tokenizer and generative models.

---

### Meta-Review · Area_Chair_gDLC · 2026-01-06

**Summary:**

The paper proposes l-DeTok, a new visual tokenizer drawn from Gaussian denoising of latents. The majority of reviewers are strongly supportive of acceptance, highlighting the simplicity of the approach, its clear motivation, and strong experimental results. AC agrees with these assessments, and recommends acceptance as well.

**Reviewer Concerns:**

The major concerns were primarily raised from Reviewer 71cM, including the lack of theoretical analysis, positioning issue, and additional comparisons with latest VAEs. The authors have clarified these points and conducted additional experiments, and AC believes that their concerns have been adequately addressed.

**Reviewer Scores:**

- Reviewer 71cM: Initially 4. The score would likely increase to 6 or above.
- Reviewer BHYG: Initially 8. Would keep his/her original positive score.
- Reviewer gSib: Initially 6. The score was increased to 8 before score resetting.
- Reviewer pDBk: Initially 8. The follow-up comment confirms that the original score would be maintained.

---

### Decision · Program_Chairs · 2026-01-26

Accept (Poster)